# Integrating Towed Underwater Video and Multibeam Acoustics for Marine Benthic Habitat Mapping and Fish Population Estimation

Alexander R. Ilich [1,*], Jennifer L. Brizzolara [1], Sarah E. Grasty [1], John W. Gray [1], Matthew Hommeyer [1], Chad Lembke [1], Stanley D. Locker [1], Alex Silverman [1], Theodore S. Switzer [2], Abigail Vivlamore [1] and Steven A. Murawski [1]

[1] College of Marine Science, University of South Florida, 140 7th Ave S, St. Petersburg, FL 33701, USA; jlbrizzolara@mail.usf.edu (J.L.B.); grastys@usf.edu (S.E.G.); jwgray@mail.usf.edu (J.W.G.); mhommeyer@usf.edu (M.H.); clembke@usf.edu (C.L.); stan@usf.edu (S.D.L.); asilverman@usf.edu (A.S.); avivlamore@usf.edu (A.V.); smurawski@usf.edu (S.A.M.)

[2] Florida Fish and Wildlife Conservation Commission, Fish and Wildlife Research Institute, 100 8th Ave SE, St. Petersburg, FL 33701, USA; Ted.Switzer@myfwc.com

\* Correspondence: ailich@usf.edu

**Abstract:** The west Florida shelf (WFS; Gulf of Mexico, USA) is an important area for commercial and recreational fishing, yet much of it remains unmapped and unexplored, hindering effective monitoring of fish stocks. The goals of this study were to map the habitat at an intensively fished area on the WFS known as "The Elbow", assess the differences in fish communities among different habitat types, and estimate the abundance of each fish taxa within the study area. High-resolution multibeam bathymetric and backscatter data were combined with high-definition (HD) video data collected from a near-bottom towed vehicle to characterize benthic habitat as well as identify and enumerate fishes. Two semi-automated statistical classifiers were implemented for obtaining substrate maps. The supervised classification (random forest) performed significantly better ($p = 0.001$; $\alpha = 0.05$) than the unsupervised classification (k-means clustering). Additionally, we found it was important to include predictors at a range of spatial scales. Significant differences were found in the fish community composition among the different habitat types, with both substrate and vertical relief found to be important with rock substrate and higher relief areas generally associated with greater fish density. Our results are consistent with the idea that offshore hard-bottom habitats, particularly those of higher vertical relief, serve as "essential fish habitat", as these rocky habitats account for just 4% of the study area but 65% of the estimated total fish abundance. However, sand contributes 35% to total fish abundance despite comparably low densities due to its large area, indicating the importance of including these habitats in estimates of abundance as well. This work demonstrates the utility of combining towed underwater video sampling and multibeam echosounder maps for habitat mapping and estimation of fish abundance.

**Keywords:** benthic habitat mapping; multibeam; fish community; underwater video

## 1. Introduction

Mapping of benthic habitats has become a critical element of living marine resource management globally [1–4]. Detailed habitat maps, at spatial scales relevant to management actions, provide the basis for protecting sensitive biota that may be vulnerable to disruptive human activities [5] and for evaluating the relationships of abundance and community structure of plants and animals to particular habitat types. Marine spatial planning, an element of ecosystem-based management, requires detailed, georeferenced information on benthic sediment types, and ecological processes within them [1,3,5,6]. In order to meet the demand for broad scale, accurate, and timely habitat mapping products, a variety

of technologies and protocols have been developed and tested [7–9]. However, most of these technologies do not individually provide comprehensive, synoptic, or relatively unambiguous interpretations of habitat features and their biotic attributes.

Benthic habitat mapping generally involves performing an acoustic survey (e.g., with a multibeam echosounder) over a region of interest with systematic transects of data derived via "mowing the lawn" [10,11]. For a multibeam echosounder (MBES), the width of individual transects and thus the efficiency of such sequential approaches are dictated primarily by water depth. MBES provides a detailed topography (bathymetry) product plus a backscatter image which measures the intensity of the returning sound pulse and can be indicative of sediment grain size, composition, and substrate type [10–16]. In addition, ground-truthing data, for example, collected from camera imagery, must be collected in order to inform or verify interpretations of the acoustic returns and understand what sediment and other geological features they represent and to identify the biota associated with these habitats [7,8,17]. Additionally, with the increasing volume of data and the greater demand for habitat maps for ecosystem-based management and marine spatial planning, scientists and managers must consider the reliability of MBES data and their derivative metrics as surrogates to classify habitat types over differing spatial scales, especially where no independent ground-truthing exists [9,17–21]. For bathymetry, these derivative features include terrain attributes such as slope, rugosity, aspect, etc. [21,22], and for backscatter, these include texture metrics such as those derived from a gray level co-occurrence matrix (GLCM) [23–29].

Benthic habitat maps have the potential to make significant contributions to fisheries science. For example, habitat maps in the Florida Keys (USA) have facilitated habitat-stratified surveys, leading to more precise and cost-effective fisheries-independent monitoring surveys [30], which have become more important for monitoring as stricter management regulations have reduced the data available from fisheries-dependent sources [31–34]. Although the west Florida shelf (WFS; Gulf of Mexico, USA) is an important commercial and recreational fishing area, much of the WFS remains unmapped and unexplored, hindering effective monitoring of fish stocks [34–36]. Additionally, fisheries are typically managed using relative indices of abundance; however, what is of interest to managers are absolute estimates of abundance [37–39] when species are managed under an overall catch quota.

In this study, we demonstrate how MBES and towed underwater video can be used in tandem to extrapolate a small video sub-sample to provide predicted habitat maps for the entire study area and estimate absolute fish abundance in a popular fishing area on the WFS known as "The Elbow". We compare the performance of two different approaches for mapping habitat: a supervised approach (random forest) and an unsupervised approach (k-means clustering) [10]. Additionally, we investigate the importance of predictors derived from different sources (bathymetry vs. backscatter) and over several spatial scales of analysis in the mapping procedure. Moreover, we test if the fish communities differ significantly among habitat types and identify the taxa driving any differences. Lastly, we demonstrate how these habitat maps can be combined with fish density data from a towed video survey to estimate fish abundance.

## 2. Materials and Methods

### 2.1. Study Area

The Elbow is hypothesized to be an ancient sea level stand shaped by wave action approximately 12,000 years ago [36,40]. The area lies 145 km west-northwest of Tampa Bay, Florida (Figure 1). It contains both hard-bottom (rock) and soft-bottom (sand) habitats, supporting associated benthic invertebrate assemblages including sponges, gorgonians, and sea urchins as well as a diverse community of reef fishes. The portion of the Elbow surveyed in this study (see next section) ranged in depth from 45 m to 65 m and contains a linear ridge that runs north to south for 16 km (Figure 1a). This area was chosen as a candidate to test and develop this analysis protocol for integrating towed video and

MBES data because the hard-bottom ridges formed by paleo-shorelines in this area are representative of what we would expect elsewhere on the WFS [41]. Additionally, vessel monitoring systems (i.e., satellite tracking) data have identified the Elbow as a reef fishing "hot spot" [42,43], and this area is of interest to fisheries management [34,36].

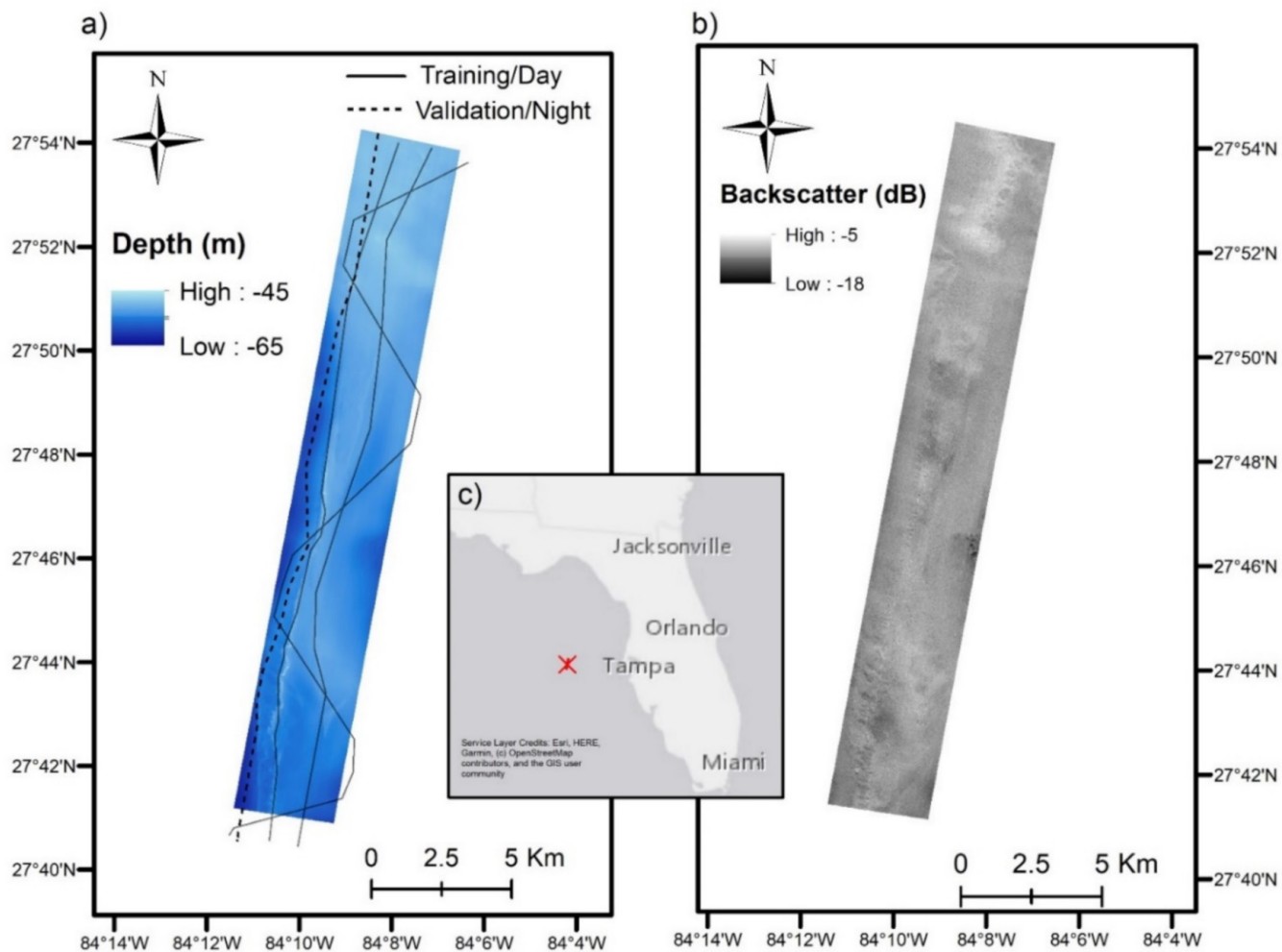

**Figure 1.** Bathymetry (**a**) and backscatter (**b**) surfaces of the Elbow on the west Florida shelf (WFS) aligned to a matching 10 m × 10 m resolution grid. The inset map (**c**) shows the location of the area relative to the state of Florida. Towed video transects overlain on the bathymetry surface with the three solid lines represent daytime transects that were used for training habitat models and in conducting fish analyses. The dashed line represents a night transect that was used for validation of habitat models and was excluded from fish analyses.

### 2.2. Data Collection and Processing

#### 2.2.1. Multibeam Echosounder Data

In December of 2015, 88 km$^2$ of the Elbow region was surveyed using a Teledyne Reson SeaBat 7125 MBES (Figure 1). This echosounder has 512 overlapping beams and was operated at 400 kHz with a 140° swath. The SeaBat 7125 was pole-mounted on the port side of the R/V *Bellows* operated by the Florida Institute of Oceanography. Navigation and motion compensation data were collected with an Applanix POS MV OceanMaster system [44]. An AML Oceanographic Micro•X was used to correct for sound launch velocity at the sonar head, and an AML Oceanographic Minos•X with an SV•Xchange sound velocity sensor was used for sound velocity profile correction to compensate for changes in the speed of sound throughout the water column.

Bathymetry data were post-processed using Caris HIPS and SIPS 10.2 with depth uncertainties less than 0.35 m [45]. The backscatter mosaic was created using the Caris SIPS

time series algorithm. The bathymetry surface and the backscatter mosaic were exported to 2 m × 2 m resolution and 1 m × 1 m resolution raster grids, respectively. The bathymetry surface was then aggregated to a coarser 10 m × 10 m resolution grid, and the backscatter was aligned to a matching grid using bilinear interpolation so that both surfaces were referenced to a common grid as required for use in the statistical substrate habitat models (Figure 1). A resolution of 10 m × 10 m was chosen, as this approximates the scale of video observations, reduces artifacts in terrain attributes, and reduces potential errors related to positional uncertainty of the towed video system.

### 2.2.2. Towed Underwater Video

The Camera-Based Assessment Survey System (C-BASS) is a towed underwater camera system custom built to non-lethally sample demersal reef fish and classify bottom types [46,47]. The C-BASS is towed behind a research vessel at speeds of 1.5–2 m s$^{-1}$ and between 2–4 m above the seafloor [46]. The system consists of four LED lights and six underwater video cameras [46,47]. All cameras on the C-BASS are oriented obliquely at a downward angle from the main horizontal chassis, as this increases the area observed, increases fish detection probability, and provides a perspective that aids in the simultaneous identification of fish species and associated habitat characteristics [47,48]. A forward-facing monochrome HD camera was used as the primary camera to identify fish and habitat types in this study, as it consistently provided the clearest imagery, however, other color onboard cameras were used to aid in identification of fish and habitats. This primary camera is a FLIR Blackfly Gige Vision camera with a Sony IMX249 image sensor (13.3 mm diagonal; 1920 × 1200 pixel resolution) and a 2/3″ Kowa LM5JC10M lens (focal length = 5 mm). This camera was mounted at an angle of 32.8° down from the main horizontal chassis. In addition to the six cameras, the C-BASS has various onboard sensors which continually record data along transects at a frequency of 1 Hz or greater, including a three-axis compass to record pitch, roll, and heading of the towbody, an altimeter to record height above the seafloor, and a CTD and fluorometer to record depth and ambient water properties [46,47].

Video transects were planned by inspection of the MBES bathymetry to maximize the likelihood of encountering a diversity of possible benthic habitat types. Four representative video transects from a February 2016 cruise aboard the R/V *Weatherbird II* were analyzed (Figure 1a). These transects consist of 15 h of video and covered 109 linear km and an area of almost 1 km$^2$, or just over 1% of the total study area. One transect followed the main north-south ridge found in the area; a second transect zigzagged across the entire study area, crossing over the main ridge multiple times to sample a broad range of habitats, and a third transect bisected the study area from north to south. These three video transects were collected during the day and were used as the training data set for creating substrate maps and were used in fish community and abundance analyses (Figure 1a). A fourth transect was collected at night and followed a smaller ridge west of the main ridge. This transect was reserved as an independent validation transect used for accuracy assessment of the predicted substrate maps. Additionally, it was excluded from fish analyses (Figure 1a), as fish identification is more difficult at night, and it would confound comparisons of fish communities across habitat types, as day and night fish communities have been shown to differ on the WFS [49].

Substrate was classified from still images extracted every 15 s, resulting in the classification of 3680 images; however, scrolling a few seconds in each direction was allowed to provide context and ensure that the classification adequately characterized the area. Images were classified according to the substrate and biotic components of the Coastal and Marine Ecological Classification Standard (CMECS) scheme [50]. Vertical relief was qualitatively assessed. All observations were binned into four broad habitat types: sand, low relief rock, moderate relief rock, and high relief rock. Low relief rock ranged from rocky habitats covered by a sand veneer to those exposed but showing little change in elevation; moderate relief was defined as a noticeable or step-like change in elevation; high relief occurred when there were large and sudden changes in elevation (Figure 2). These

relief categories are qualitative and were assessed visually from the video imagery but correspond well with those described by Smith et al. [30] for mapping the Florida Keys (Figure A1). The biotic elements were not included in the final broad categorization, as the main observable biotic features were attached fauna such as sponges and corals, and they largely tracked where there was rock; however, the biotic component did prove particularly useful in identifying areas where low relief rock was covered by a thin sand veneer, as the presence of attached fauna such as sponges and corals indicates the likely presence of hard substrate beneath the sand as attachment for these organisms [51]. In the cases of mixed habitat classes, areas were considered to be rock where a thin sand veneer was overlain on rock or where large moderate to high relief rocky features were exposed. Conversely, areas characterized by a few rubble piles or very small, isolated, low relief rocky features within a larger expanse of sand were considered to be sand.

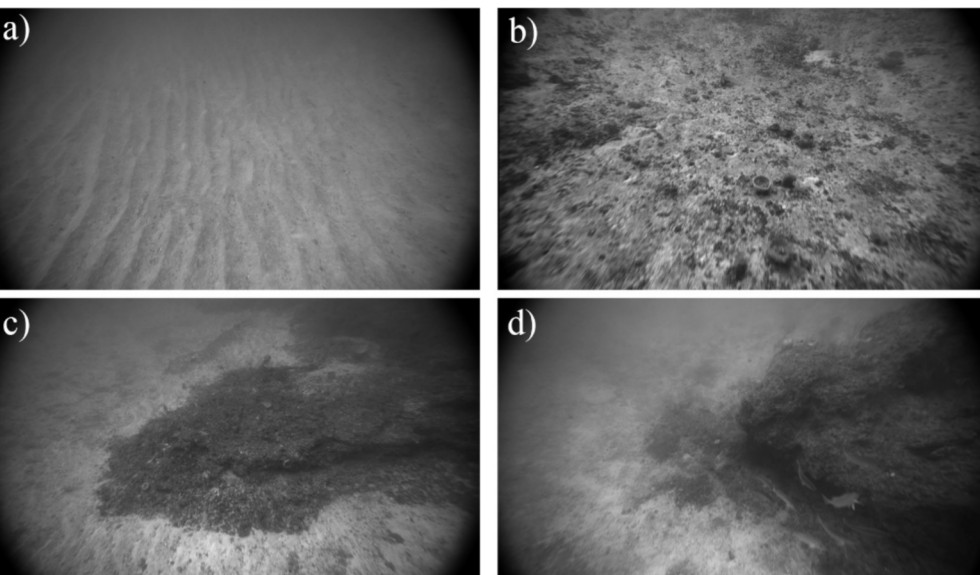

**Figure 2.** Substrate types observed in the Elbow of the west Florida shelf include sand (**a**) and rock substrate of varying levels of relief: low (**b**), moderate (**c**), and high (**d**).

For fish counts, all videos were analyzed manually using the CVision fish counting software [52]. Fish were only counted if they were observable in the primary camera, though other cameras were used to aid species identification. Fish were identified to the lowest taxonomic level possible. If fish could not be identified to species, they were identified to a higher taxonomic level (e.g., genus or family). The C-BASS is best suited for surveying larger-bodied reef fish, such as groupers and snappers, and previous work has shown that fishes smaller than approximately 15–20 cm (i.e., average length of most adult squirrelfish and bigeye species in the Gulf of Mexico) cannot be reliably identified unless they have very distinguishing features given speed, elevation, and orientation of the camera towbody. Fishes smaller than the average adult damselfish (approximately 10 cm) are virtually undetectable on C-BASS imagery [53]. Any individuals observed but otherwise unidentifiable due to visibility, behavior, or otherwise were categorized as large or small unidentified fish.

### 2.3. Predictive Habitat Mapping

2.3.1. Response Variable: Ground-Truth Substrate Observations

Both supervised (random forest) and unsupervised (*k*-means clustering) statistical models were fit to predict seafloor substrate (rock vs. sand) based on the MBES data and their derivative features [10]. Ground-truth substrate classifications from the towed video were used as the response variable, and bathymetry, backscatter, and their derivative features were the predictors (independent variables) in the predictive statistical models.

Due to a comparatively small number of observations for higher relief rocky habitats and their relatively small spatial size relative to the positional error associated with our towed system, these models did not attempt to predict vertical relief but rather substrate type. To reduce the influence of spatial autocorrelation on accuracy assessment, ground-truth observations from three transects were used to train the model, and one entire transect was reserved for the accuracy assessment of predictions (Figure 1). To reduce the effect of positional uncertainty of the video system and confusion due to mixed habitats or habitat boundaries [54–57], only habitat observations that were the same as their previous and subsequent observations were used to fit and assess the models.

2.3.2. Predictor Variables: MBES Data and Their Derivative Features

The 10 × 10 m resolution bathymetry and backscatter grids were used to calculate various derivative metrics (Table 1). These metrics include terrain attributes derived from the bathymetry surface [21,22] and texture measures derived from the backscatter mosaic using GLCMs [23,24] as well as the local mean and the standard deviation of backscatter using varying window sizes. The 10 × 10 m bathymetry and backscatter surfaces along with their derivative features were used as the predictor variables in the statistical models.

**Table 1.** Derivative features from the 10 m × 10 m bathymetry and backscatter surfaces for the Elbow of the west Florida shelf. Features were calculated using eight different scales of analysis from a 3 × 3 pixel to a 69 × 69 pixel moving window, and the resulting surfaces all have 10 m × 10 m resolution. Formulas for texture metrics are from Hall-Beyer [24]. N = number of rows or columns in the gray level co-occurrence matrix (GLCM) (equal to the number of gray levels, 32); i = row indices of the GLCM (equal to gray level of reference pixel); j = column indices of the GLCM (equal to gray level of neighboring pixel); $P_{i,j}$ = probability (relative frequency) of neighboring pixels having gray levels i and j.

| Feature | Source | Description/Algorithm |
|---|---|---|
| Local mean of bathymetry | Bathymetry | Mean of bathymetry in a given window |
| Local standard deviation of bathymetry | Bathymetry | Standard deviation of bathymetry in a given window (a measure of rugosity) |
| Eastness | Bathymetry | $\sin(aspect)$ |
| Northness | Bathymetry | $\cos(aspect)$ |
| Slope | Bathymetry | Measure of the rate of change in bathymetry |
| Topographic Position Index | Bathymetry | Indicates whether a location is a local high or low |
| Local mean of backscatter | Backscatter | Mean of backscatter in a given window |
| Local standard deviation of backscatter | Backscatter | Standard deviation of backscatter in a given window (a measure of heterogeneity) |
| GLCM Mean (μ) | Backscatter | $\sum\limits_{i,j=0}^{N-1} i\left(P_{i,j}\right)$ |
| GLCM Variance (σ²) | Backscatter | $\sum\limits_{i,j=0}^{N-1} P_{i,j}(1-\mu_i)^2$ |
| GLCM Homogeneity | Backscatter | $\sum\limits_{i,j=0}^{N-1} \frac{P_{i,j}}{1+(i-j)^2}$ |
| GLCM Contrast | Backscatter | $\sum\limits_{i,j=0}^{N-1} P_{i,j}(i-j)^2$ |
| GLCM Dissimilarity | Backscatter | $\sum\limits_{i,j=0}^{N-1} P_{i,j}|i-j|$ |
| GLCM Entropy | Backscatter | $\sum\limits_{i,j=0}^{N-1} P_{i,j}\left[-\ln\left(P_{i,j}\right)\right]$ |
| GLCM Angular Second Moment | Backscatter | $\sum\limits_{i,j=0}^{N-1} P_{i,j}^2$ |
| GLCM Correlation | Backscatter | $\sum\limits_{i,j=0}^{N-1} P_{i,j}\frac{(i-\mu)(j-\mu)}{\sigma^2}$ |

Bathymetric terrain attributes and backscatter texture metrics were computed at multiple scales of analysis by varying the window sizes over which the metrics were

calculated [22]. Including predictors derived at a range of different scales is considered "best practice" in benthic habitat mapping [22,25,56,58,59]. Systematically representing multiple scales was accomplished by varying the size of the analysis window according to the Fibonacci sequence [22]. The Fibonacci sequence was used to determine the radius in number of pixels around a central pixel for eight different scales of analysis. This resulted in derivative features being calculated at eight different window sizes from $3 \times 3$ pixels to $69 \times 69$ pixels ($3 \times 3$, $5 \times 5$, $7 \times 7$, $11 \times 11$, $17 \times 17$, $27 \times 27$, $43 \times 43$, $69 \times 69$; scale factors ranging from 30–690 m).

The bathymetric terrain attributes were calculated in the R programming language version 3.6.3 [60] using the raster package [61]. The terrain attributes chosen were slope, eastness and northness components of aspect (i.e., the orientation of the slope), local standard deviation of bathymetry (a measure of rugosity), local mean of bathymetry, and topographic position index (e.g., a local high or low [62]). These attributes were selected based on suggestions from Lecours et al. [21]; however, topographic position index (TPI) was substituted for relative difference from mean value (RDMV), as they both are used to determine local highs and lows, but TPI has a more intuitive meaning, and the denominator in RDMV can cause the metric to be undefined, leading to voids in the resulting surface. Horn's method [63] was used to derive slope and aspect, as this is the most widely used and widely available algorithm for computing these metrics [21]; however, its calculation is restricted to a $3 \times 3$ window. To extend these metrics (slope, eastness, and northness) to multiple scales, these attributes were first derived (referred to as native scale), and then the local mean of each attribute was calculated using varying window sizes (method three in Dolan [64]).

For the backscatter derivatives, the local mean and the local standard deviation of backscatter intensity were calculated across the range of window sizes. Additionally, texture measures based on GLCMs were calculated [23,24]. Texture metrics used were GLCM contrast, GLCM dissimilarity, GLCM homogeneity, GLCM angular second momentum (ASM), GLCM entropy, GLCM mean, GLCM variance, and GLCM correlation [24]. These texture metrics can be broken down into three groups: the contrast group (contrast, dissimilarity, and homogeneity), the orderliness group (ASM and entropy), and the descriptive statistics group (mean, variance, and correlation) [24]. For the GLCM texture metrics, the 10 m x 10 m backscatter surface was first quantized to 32 gray levels (discrete integer values ranging from 0–31) using equal probability quantization [23,65]. Then, a symmetrical GLCM was tabulated in all four directions ($0°/180°$, $45°/225°$, $90°/270°$, $135°/315°$, where $0°$ is directly to the right and degrees increase counter-clockwise), and the value of texture metrics was averaged over all directions to get directionally/rotationally invariant measures of texture. Although there are many software packages to calculate these GLCM textures, many are proprietary, and there can be inconsistencies in the results among different software [24,66]. In some cases, these inconsistencies have prevented their use in benthic habitat mapping studies [67]. For this study, free open source software was developed to calculate these texture metrics on raster data and is available as an R package [68].

### 2.3.3. Estimating Towed Video Position

To associate ground-truth habitat observations from the towed video with the relevant MBES data, the position of the video at a given time must be estimated. This was done trigonometrically [15,48,69]. The layback of the of the towed video system behind the vessel was calculated using Pythagorean theorem based on the cable out from the winch, the depth of the C-BASS, the vertical offset of the height of the A-frame block relative to the waterline, and the fore/aft offset between the A-frame and the GPS antenna (Equation (1) and Figure 3). The layback was converted to a time delay by dividing it by the average

speed over the preceding minute and then was used to assign C-BASS a position by assuming that the C-BASS exactly followed the ship track but lagged in time.

$$\text{Layback} = y + \sqrt{C^2 - (z + A)^2} \tag{1}$$

$C:$ cable out $(m)$; $z:$ CBASS Depth $(m)$; $A:$ $A -$ Frame height $(m)$; $y:$ fore/aft offset.

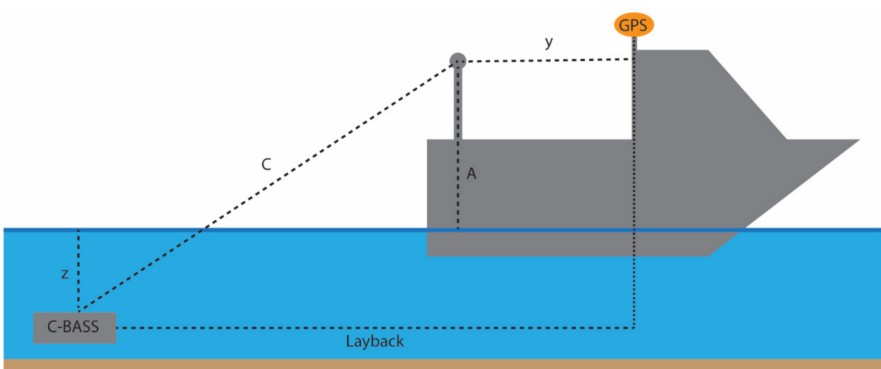

**Figure 3.** Schematic representation showing the physical meaning of the parameters from Equation (1) that were used to calculate the layback of the Camera-Based Assessment Survey System (C-BASS) system behind the ship's GPS antenna.

2.3.4. Classification Algorithms

Both supervised and unsupervised classification procedures were employed to predict substrate over the full area of the MBES survey [10]. The supervised classification map was created using a balanced random forest algorithm with down-sampling [70,71] via the RSToolbox [57,72] and ranger [73] R packages. The random forest algorithm has been implemented in a number of previous benthic habitat mapping studies [20,25,26,56,74–76] and has been found to have good performance for mapping benthic habitats when compared to other classifiers [74,77]. The random forest algorithm works much like a traditional decision tree where each node determines the optimal split in the predictor variables to best separate groups, however, rather than simply fitting one decision tree, a "forest" of many decision trees (e.g., hundreds to thousands) is fit to the data based on a bootstrap sample (sampling with replacement) of the data, and each node on a tree is only given access to a random subset of predictors [70,78]. Due to class imbalance (many more observations of sand as compared to rock), we implemented a balanced random forest algorithm using down-sampling, meaning that each bootstrap sample used to fit an individual decision tree contained an equal number of observations of each class, and the number of observations of each class was equal to the number of observations in the minority class. This has been found to be an effective way to improve predictions using unbalanced data [71]. The number of decision trees was set to 3000, and the plot of "out-of-bag" (OOB) error vs. the number of trees was used to verify that a sufficient number of trees was used by ensuring that the error rate reached an asymptote, and the number of variables available at each split was tuned to maximize Kappa using five-fold cross validation. Cohen's Kappa ($\kappa$) is a measure of the level of agreement between two sets (often between predictions and observations). The value of Kappa is equal to one if there is complete agreement between the two sets, zero if the agreement is no greater than what could occur by chance, and negative if the agreement is less than what could occur by chance. Some rules of thumb for Kappa are that values less than zero represent poor agreement, from 0–0.2 represents slight agreement, from 0.2–0.4 represents moderate agreement, from 0.6–0.8 represents substantial agreement, and values from 0.8–1 represent almost perfect agreement [79,80].

The unsupervised statistical model used was *k*-means clustering [81], implemented via the RSToolbox R package [57,72]. *K*-means clustering is one of the most used clustering algorithms for both terrestrial and marine mapping applications [56,57] and is widely

available in many different software environments. As the *k*-means clustering algorithm requires the number of clusters to be specified a priori, a rule of thumb is to use twice the number of the desired classes [82]. Since the desired thematic resolution was a two-class map of rock vs. sand, the *k*-means algorithm was run using four clusters. The ground-truth habitat points from the training data set were then used post-hoc to interpret the acoustic clusters. Each cluster was interpreted as a substrate type based on majority vote of all ground-truth habitat points from the training set contained within that cluster.

### 2.3.5. Variable Selection and Dimensionality Reduction

Calculating the derivative features for bathymetry and backscatter across multiple scales resulted in 130 predictor variables (Table 1). GLCM correlation was only included at five different scales ($3 \times 3$, $5 \times 5$, and $7 \times 7$ were removed), as the metric is undefined if all values in the window have the same value, which is more likely for smaller window sizes and leads to voids in the resulting surfaces. Of these 130 predictors, many were redundant and highly correlated, thus it was necessary to conduct variable selection and dimension reduction [56].

For the supervised classification model, the Boruta selection algorithm was used to determine important predictors [77,83]. This method fits many random forest models and uses an iterative process to determine if a predictor variable has an importance score significantly greater than a permuted version of itself. The significance threshold was set at $\alpha = 0.05$, and variable importance was calculated as the unscaled permutation-based mean decrease in accuracy of OOB observations [84,85]. To further reduce the number of predictors and remove high co-linearity predictors, the Pearson correlation was calculated between all remaining predictors, and a predictor was removed from the model if it was highly correlated ($|r| > 0.8$) with a predictor that had a higher average importance score as determined by the Boruta procedure [25,56,58]. This is similar to procedures employed in other studies (e.g., [25,58]) however, in our study, rather than including only the most important scale of a given predictor, the same predictor was allowed to be included at multiple scales based on the logic that a given predictor may be important at several different scales and may be representative of different processes, especially if those different scales are not highly correlated.

For the unsupervised classification model, all predictors (multibeam bathymetry, backscatter, and their derivative features) were *z*-score normalized to minimize the effects of differing ranges and units among predictors, and a principal components analysis (PCA) was conducted to remove the effect of multi-collinearity [56,86]. To remove redundant principal components (PCs), only a subset of the original PCs was retained. A PC was only retained if it explained more variance than what would be expected if the total variance was divided randomly amongst all the PCs as modeled by a broken-stick distribution [87–89]. The retained PCs were then used in the *k*-means clustering procedure [81].

### 2.3.6. Accuracy Assessment

Both supervised and unsupervised substrate maps were assessed using the validation transect in terms of overall, producer's, and user's accuracy as well as the Cohen's Kappa statistic [90]. User's and producer's accuracy statistics are useful for assessing errors of omission (false negatives) and errors of commission (false positives) for each substrate type. The Kappa scores of the two maps were then compared using a Monte-Carlo permutation procedure with 999 iterations to test whether the supervised classification map had significantly greater performance ($\alpha = 0.05$) than the unsupervised classification map [91].

An entropy map, which displays the uncertainty in predicted substrate for each individual pixel, was generated for the supervised classification map. Entropy was calculated

based on the proportion of decision trees in the model that voted for each class within a given pixel using the Shannon entropy formula (Equation (2)) [57,92,93].

$$\text{Entropy} = -\sum_{i=1}^{M} [p_i * \ln(p_i)] \tag{2}$$

$p_i$ : probability that a cell is of class i; i : class number; M : number of classes.

### 2.3.7. Map Comparison

To compare maps, a difference map was created to visually depict where and how classifications from each model agreed and disagreed [94]. In addition to comparing observations to predictions for accuracy assessment, Kappa can also be used to compare how well two maps agree with each other. Moreover, since, when used in the spatial context, Kappa confounds similarity in location with similarity in quantity, we can decompose Kappa into Klocation and Khisto where Kappa is the product of these two terms [80,95,96]. Klocation quantifies how well the spatial allocation of the classes agree between the two maps, and Khisto quantifies the agreement between the fraction of pixels assigned to each class [80,95,96]. As such, Kappa, Klocation, and Khisto were calculated to compare the supervised and the unsupervised classification maps.

### 2.3.8. Vertical Relief

In addition to substrate, vertical relief was determined, as this has been found to be an important factor relating to fish community composition and abundance in many studies [34,53,97–101]. Vertical relief was calculated directly from the 10 m bathymetry surface using a moving 3 × 3 pixel moving window by taking the depth value of the central pixel in the window and subtracting the minimum depth of the surrounding pixels. These relief values were then reclassified into categories (low, moderate, and high relief) using the thresholds determined by Smith et al. [30] for the Florida Keys (low relief <1 m, moderate relief >1 and ≤2 m, and high relief >2 m) as these previously established thresholds corresponded well with our observations based on changes in total depth over 15 s bins as calculated from C-BASS' onboard sensors (Figure A1). This was then layered over the best performing substrate map to delineate rock habitats of various types of relief, resulting in a final habitat map with four classes: sand, low relief rock, moderate relief rock, and high relief rock.

### 2.4. Fish Community Analyses and Abundance Estimates

#### 2.4.1. Fish Densities

Using the fish counts and the corresponding habitat classifications from the video data, a species-by-site matrix of fish counts and habitat observation binned to 15 s intervals was created by associating each individual fish with the closest habitat observation in time. For each 15 s bin, fish counts were converted to densities by dividing fish counts by the cumulative area viewed in the video over the 15 s observation window (Equation (3)). The horizontal angle of view of the system in air was calculated based on the properties of the camera system [102]. This angle of view was then adjusted for the refraction of light in seawater using Snell's Law, and the average width of the frame for each 15 s bin was calculated using the trigonometric approach of Grasty [53], which estimates the frame width using the altitude and the pitch of the camera relative to the bottom as well as the horizontal angle of view of the camera in seawater. Distance traveled by the video system was calculated using ship speed; the area viewed was calculated by multiplying the frame width by the distance traveled. For calculations, median values of the ship speed, C-BASS altitude, and C-BASS pitch over each 15 s bin were used (see Supplementary Materials for full details).

### 2.4.2. Multivariate Community Analysis

Differences in the fish communities among the four broad habitat types (sand, low relief rock, moderate relief rock, and high relief rock) were analyzed. If the same habitat was observed multiple times sequentially along a transect, fish counts were aggregated, and the area viewed summed across all contributing 15 s bins. Counts were converted to densities by dividing the number of fishes by the area viewed (see above for area calculations).

To assess differences in fish communities among habitats, a non-parametric permutation based analysis of variance (PERMANOVA) [103] tested the null hypothesis of no significant difference in fish community composition and abundance among habitat classes. Species composition differences among individual pairs of habitat classes were evaluated using pairwise PERMANOVA tests. Prior to conducting the PERMANOVA, the validity of the assumption of homogeneity of multivariate dispersion was checked using a multivariate analogue to the Levene's test [104]. A canonical analysis of principal coordinates (CAP) was conducted to identify which taxa were responsible for driving these compositional differences [105]. These analyses were conducted in R using the vegan [106,107] and BiodiversityR [108,109] R packages. All fish counts were square-root-transformed to reduce the influence of occasional large aggregations, and the Bray–Curtis dissimilarity metric was used to calculate dissimilarity between samples [110]. Moreover, any observations where no fish were observed were removed for these analyses, as that is a requirement of the Bray–Curtis dissimilarity metric since it cannot be calculated based on joint species absences [110]. Significance was assessed at $\alpha = 0.05$, and $p$-values for pair-wise comparisons were adjusted using Holm's sequential Bonferroni procedure to account for the effect of multiple comparisons [111,112].

### 2.4.3. Fish Abundance Estimates

Based on the results of pairwise PERMANOVA tests, habitat classes were merged if their pairwise comparisons were not significantly different. The average density for each fish taxa by habitat type was calculated (Equation (4)) and 95% confidence intervals determined via bootstrap resampling with 999 iterations [113]. The combined map of substrate and vertical relief was used to calculate the total area of each habitat type. These areas were used to scale habitat-specific densities up to abundance estimates by multiplying the area of each habitat type by the habitat-specific density for a given taxa and then summing across habitat types to provide an estimate of total abundance for the study area (Equation (5)). The 95% confidence intervals for abundance were calculated by substituting the lower and the upper bounds of the habitat-specific density confidence intervals as $\overline{D}_h$ in the calculations of abundance.

$$D_b = c_b / A_b \tag{3}$$

$$\overline{D}_h = \frac{\sum_{n=1}^{n} c_b}{\sum_{n=1}^{n} A_b} = \frac{\sum_{n=1}^{n} (D_b * A_b)}{\sum_{n=1}^{n} A_b} \tag{4}$$

$$Abd = \sum_{h=1}^{nh} (\overline{D}_h * A_h) \tag{5}$$

$D_b$: Average density of fish taxa for bin b;
n: number of bins for a given habitat type;
$c_b$: number of fish of a given taxa in bin b;
$A_b$: Aera viewed by the camera over bin b;
$\overline{D}_h$: Average density of fish taxa for habitat type b;
Abd: Abundance of fish taxa summed across all habitat types;
nh: number of habitat classes;
$A_h$: Area of habitat type h in the entire study area.

## 3. Results

### 3.1. Predictive Habitat Mapping

#### 3.1.1. Preparing Ground-Truth Data

The ground-truth dataset consisted of 3680 observations where each observation was the substrate classification determined from the video at 15 s intervals. This consisted of 473 observations of rock, 3195 observations of sand, and 12 observations where the substrate was not discernable. After censoring substrate determinations that differed from their previous and subsequent observations (procedure described above), those where substrate was not visible, and observations beyond the bounds of the MBES survey, there remained 238 observations of rock and 2533 observations of sand. After splitting these data into training and validation sets, the training data set consisted of 210 observations of rock and 1947 observations of sand, and the validation data consisted of 28 observations of rock and 586 observations of sand (Figure 4).

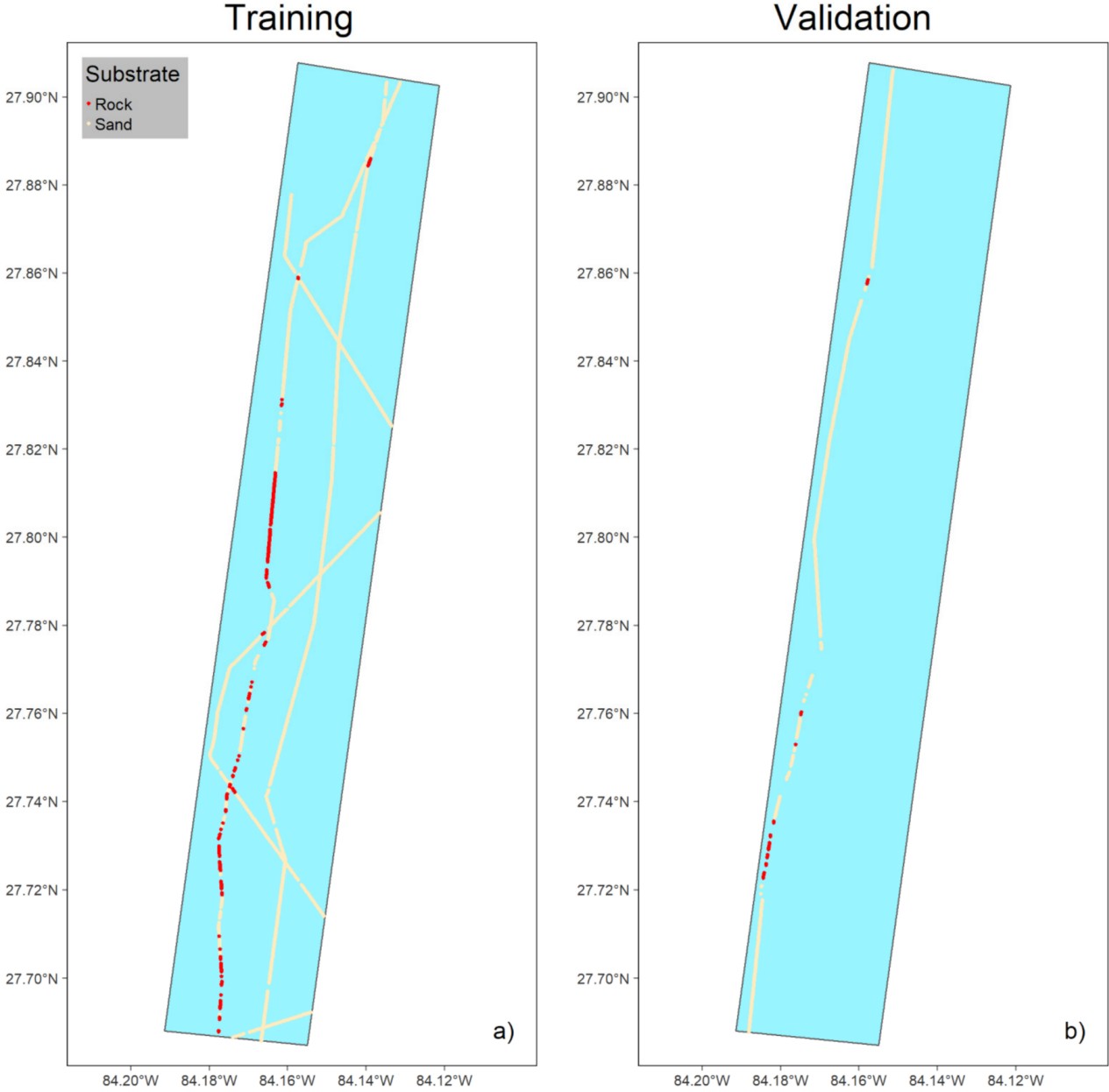

**Figure 4.** Ground-truth observations of substrate from the towed video system for training (**a**) and validation (**b**) of statistical substrate classification models. The blue area represents the multibeam echosounder (MBES) survey area.

### 3.1.2. Supervised Classification

The Boruta variable selection procedure retained 120 of the 130 predictors; however, after removing predictors that were highly correlated with a more important predictor ($|r| > 0.8$), only 12 predictor variables remained. These predictors were local mean of bathymetry ($27 \times 27$), TPI ($17 \times 17$), slope ($11 \times 11$), eastness ($7 \times 7$ and $69 \times 69$), northness (native and $11 \times 11$), local standard deviation of backscatter ($11 \times 11$ and $27 \times 27$), GLCM correlation ($11 \times 11$ and $43 \times 43$), and GLCM variance ($27 \times 27$; Figure 5). After variable selection, the optimal number of variables available at each split was determined to be four, and the number of decision trees was found to be sufficient (Figure A2). The tuned and fitted model was then used to predict substrate for the entire study area, resulting in a map consisting of 3.82 km$^2$ of rock and 83.75 km$^2$ of sand (Figure 6). Accuracy assessment was performed on the validation data; the confusion matrix with performance metrics is presented in Table 2.

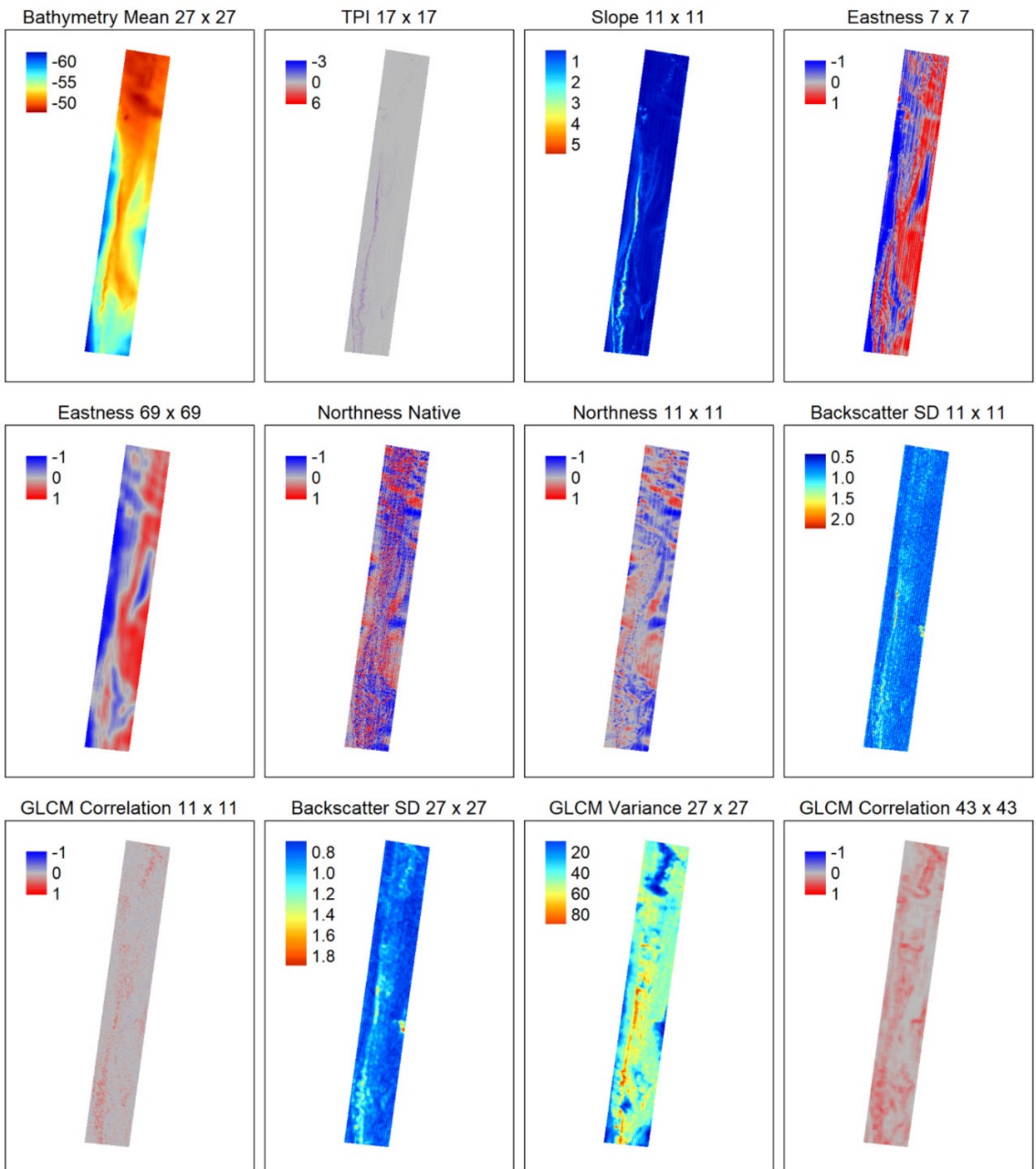

**Figure 5.** Retained predictor variables used in the random forest supervised classification model for substrate.

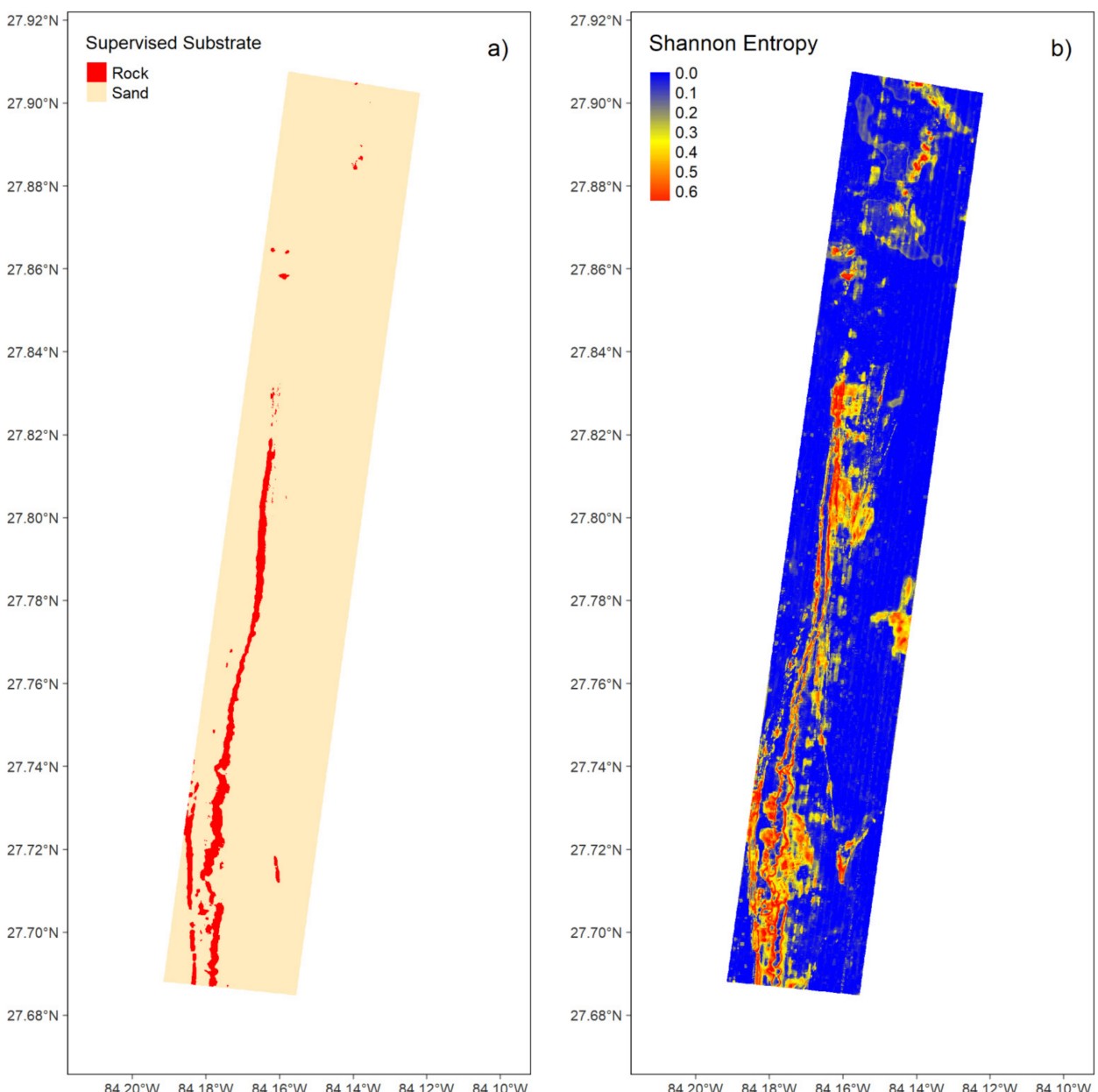

**Figure 6.** Substrate map of the Elbow of the west Florida shelf with 10 m x 10 m resolution determined through supervised (random forest) classification (**a**) and the corresponding Shannon entropy map describing the uncertainty of the classification of each pixel (**b**).

**Table 2.** Confusion matrix and user's (row-wise) accuracy, producer's (column-wise) accuracy, overall accuracy, and Cohen's Kappa statistic ($\kappa$) for the substrate map of the Elbow of the west Florida shelf derived using supervised classification.

|  | **Rock** | **Sand** | **User's Accuracy** |  |
|---|---|---|---|---|
| **Rock** | 23 | 9 | 71.9% |  |
| **Sand** | 5 | 575 | 99.1% |  |
| **Producer's Accuracy** | 82.1% | 98.5% | **Overall Accuracy** | 97.7% |
|  |  |  | $\kappa$ | 0.75 |

The variable importance is shown in Figure 7. The most important predictor was slope (11 × 11), followed by standard deviation of backscatter (27 × 27). Although some predictors were more important than others, all predictors had positive values indicating that they all provided some benefit to the model's overall predictive ability (Figure 7). Uncertainty in predicted substrate for each individual pixel is presented in the entropy map (Figure 6b).

**Figure 7.** Permutation variable importance of the MBES derived predictor variables in determining substrate type for the supervised (random forest) classification model in the Elbow of the west Florida shelf. The reported values of mean decrease in accuracy are unscaled (not divided by standard deviation) as suggested in Strobl and Zeileis [84].

### 3.1.3. Unsupervised Classification

The first nine PCs explained more variance than could be expected if the total variance was divided randomly amongst the PCs (Figure 8), and as such, they were retained for use in *k*-means clustering (Figure 9). The nine PCs explained 86% of the total variance. A *k*-means clustering procedure with four clusters was conducted on the retained PC layers (Figure 10a). Acoustically derived clusters were then assigned to substrate classes based on the majority of the ground-truth points within that cluster (Table 3 and Figure 10b). This substrate map predicts the study area to consist of 3.74 km² of rock and 83.83 km² of sand. Accuracy assessment was performed on the validation data; the confusion matrix with performance metrics is presented in Table 4.

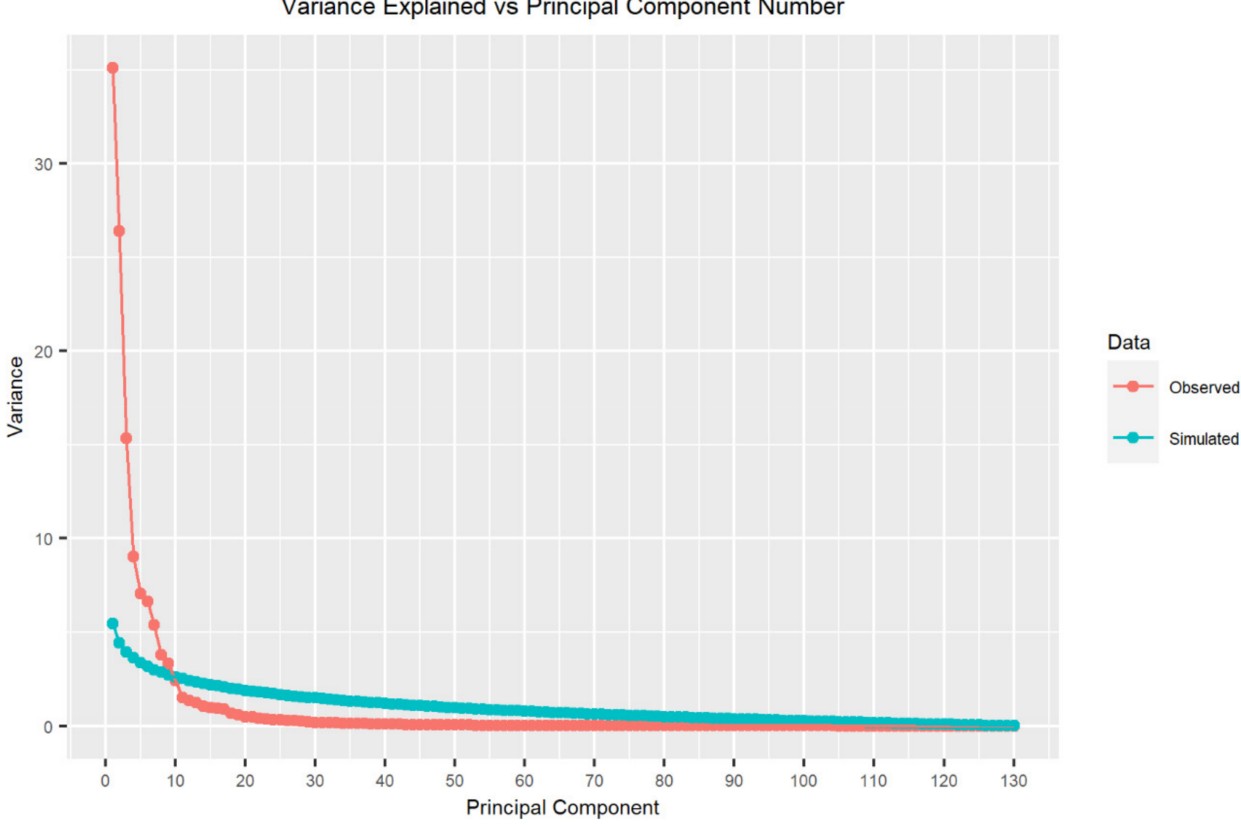

**Figure 8.** Plot of the percent of variance explained by each individual principal component for observed data and simulated data in which the variance was distributed randomly amongst the principal components as modeled by a broken-stick distribution.

**Table 3.** Substrate interpretation of the map of the four acoustic clusters of the Elbow determined through *k*-means clustering (Figure 10a) using the ground-truth habitat observations from towed video in the training set (Figure 4a). The number of ground-truth points of each substrate type within an acoustic cluster were counted, and the acoustic cluster was assigned a substrate type based on the class of the majority of the images contained within that acoustic cluster.

| Cluster | Rock Observations | Sand Observations | Assigned Substrate |
|---------|------------------|-------------------|---------------------|
| 1 | 21 | 229 | Sand |
| 2 | 179 | 75 | Rock |
| 3 | 10 | 1447 | Sand |
| 4 | 0 | 194 | Sand |

**Table 4.** Confusion matrix and user's (row-wise) accuracy, producer's (column-wise) accuracy, overall accuracy, and Cohen's Kappa statistic ($\kappa$) for the substrate map of the Elbow of the west Florida shelf derived using unsupervised classification.

| | Rock | Sand | User's Accuracy | |
|---|---|---|---|---|
| **Rock** | 10 | 0 | 100.0% | |
| **Sand** | 18 | 585 | 97.0% | |
| **Producer's Accuracy** | 35.7% | 100.0% | **Overall Accuracy** | 97.1% |
| | | | $\kappa$ | 0.51 |

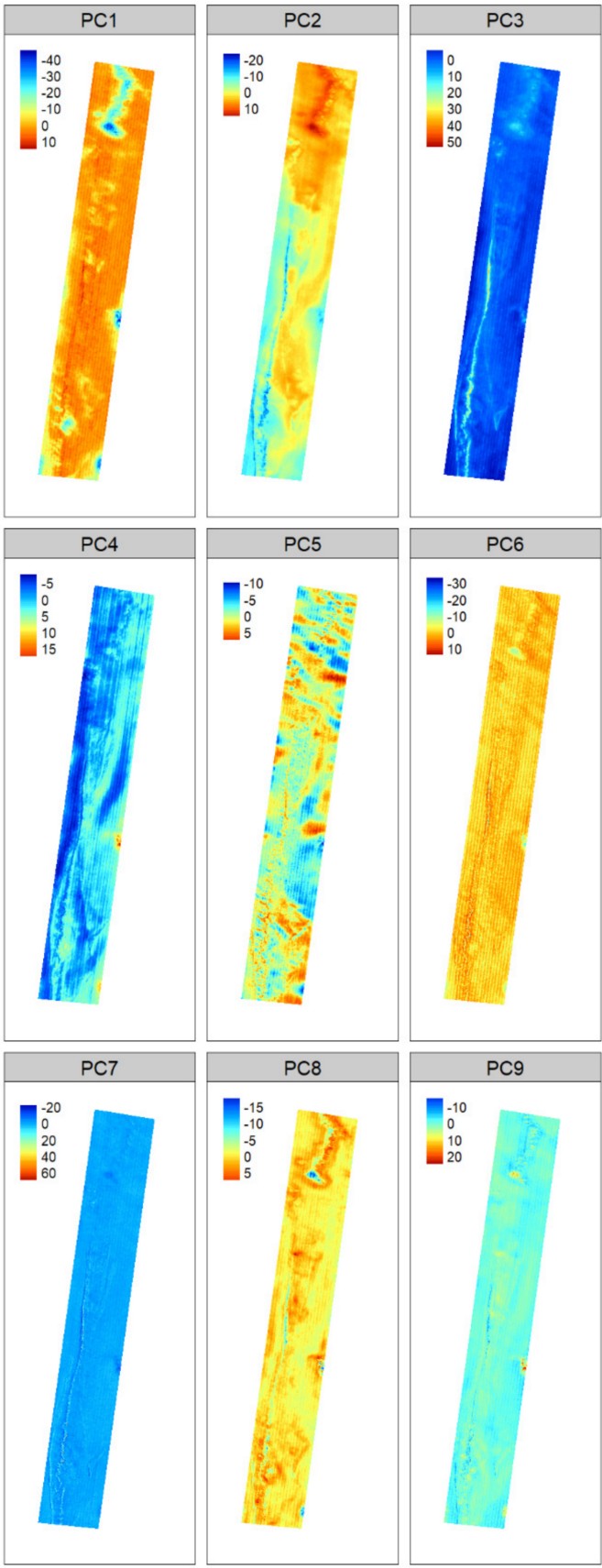

**Figure 9.** Plot of the nine retained principal components for use in the *k*-means clustering procedure.

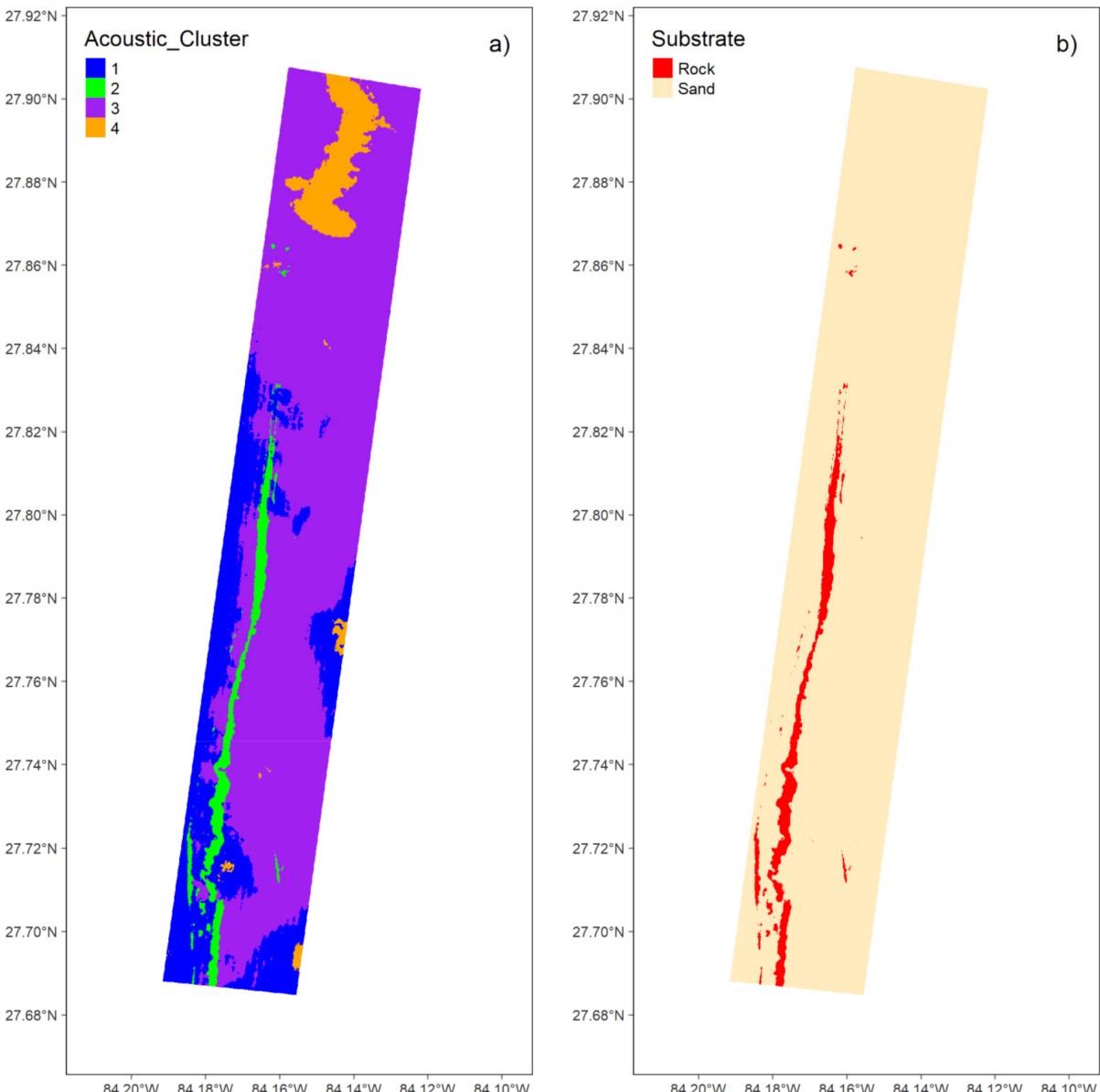

**Figure 10.** Maps for the Elbow of the west Florida shelf of the four acoustic clusters from the k-means clustering procedure of the first nine principal components of the MBES derived predictors (**a**) and the resulting unsupervised substrate map based on interpretation of the acoustic clusters from the ground-truth habitat observations from the towed video system (**b**) with 10 m × 10 m resolution.

### 3.1.4. Map Comparison

The supervised substrate map predicted 0.08 km$^2$ more rock than the unsupervised substrate map. These differences can be visualized in Figure 11. The two maps agree 98.33% of the time; the supervised classification map predicts rock where the unsupervised classification map predicts sand 0.87% of the time, and the supervised map predicts sand where the unsupervised map predicts rock 0.80% of the time. The comparison of Kappa, Klocation, and Khisto shows "almost perfect agreement" in both the spatial allocation and the relative frequency of class assignment (Klocation = 0.807 and Khisto of 0.989). Overall, this corresponds to a Kappa value (Klocation*Khisto) of 0.798, indicating a "substantial level" of agreement between the two maps. When comparing the predictions on the validation data set for accuracy assessment, the supervised classification model was found to have greater performance ($\kappa$ = 0.75 vs. $\kappa$ = 0.51). This difference was found to be

significant ($p = 0.001$) using a one-tailed Monte-Carlo permutation test, indicating that the supervised classification procedure had significantly greater predictive performance than did the unsupervised classification method.

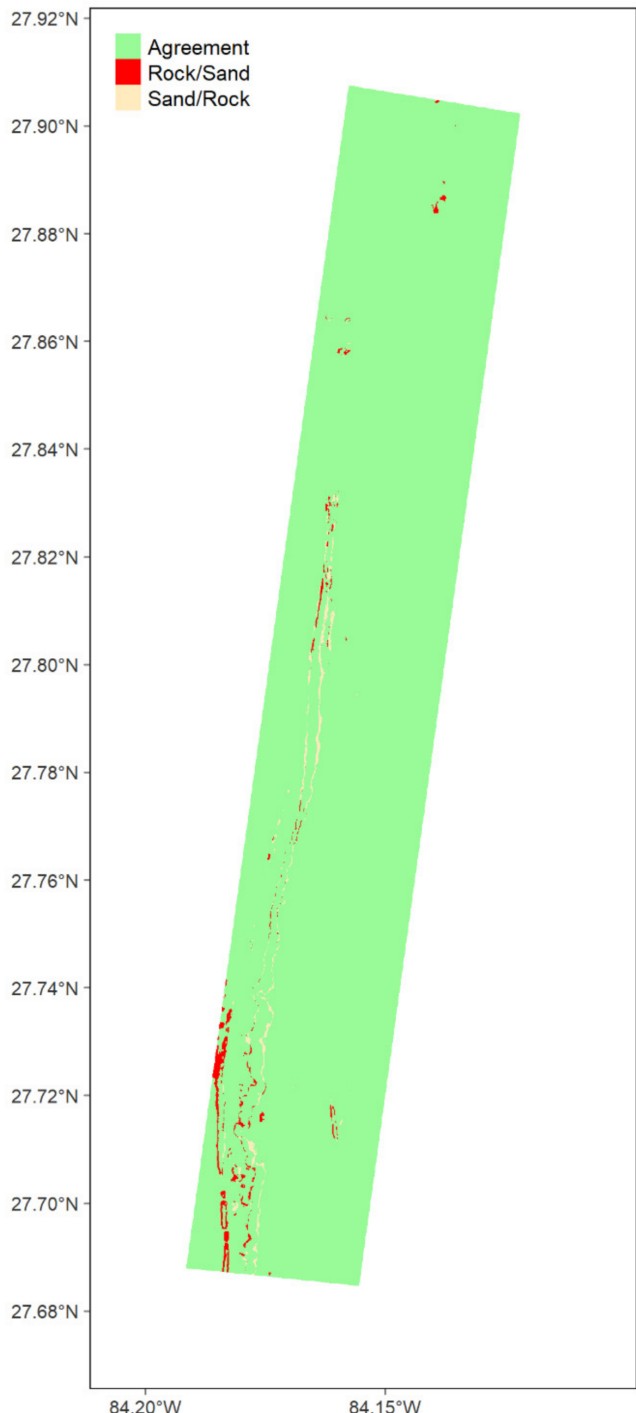

**Figure 11.** Difference map showing where the modeled predictions of substrate agreed and disagreed for the supervised and the unsupervised classification models. For disagreement, the legend is formatted with the supervised/unsupervised model prediction.

### 3.1.5. Vertical Relief

The reclassified relief map was layered onto the substrate map created using the supervised classification procedure, as it had better performance than the unsupervised

procedure, to create a "habitat map" consisting of a combination of substrate and vertical relief (Figure 12). This map predicts 83.75 km$^2$ of sand, 3.46 km$^2$ low relief rock, 0.31 km$^2$ of moderate relief rock, and 0.05 km$^2$ of high relief rock.

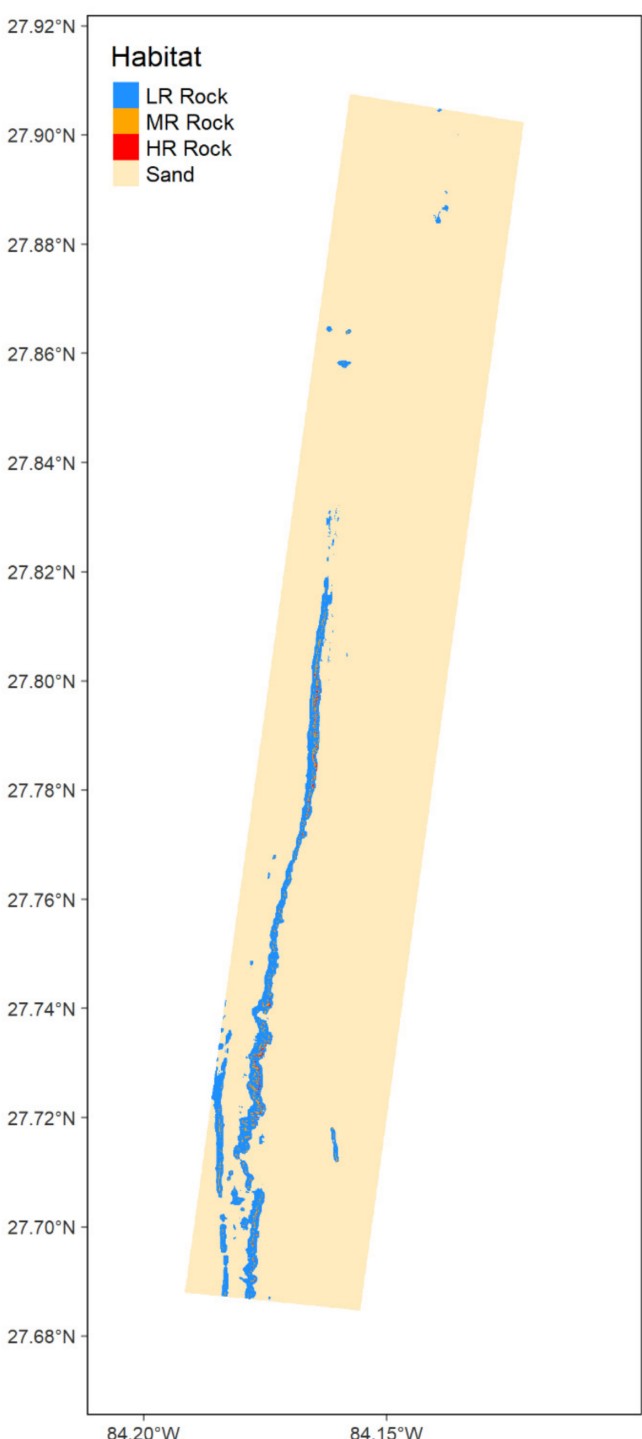

**Figure 12.** Combined substrate and vertical relief map of the Elbow of the west Florida shelf with 10 m × 10 m resolution determined through supervised (random forest) classification and maximum vertical relief calculated from the bathymetry around a central pixel. Thresholds for vertical relief were low relief <1 m, moderate relief >1 and ≤2 m, and high relief >2 m [30].

### 3.2. Fish Community Analyses and Abundance Estimates

3.2.1. Fish Densities

The average densities for each fish taxon by habitat type were calculated; however, moderate and high relief rock were merged into one category since the fish communities in them were not found to be significantly different in the pairwise comparisons (see next section). Fish densities were estimated by habitat type for 36 species or species groups as well as for all fishes combined. For most taxa (33 of the 36 species/species groups), average fish density was highest over rocky habitats, and only three species (Rainbow Runner, remora spp., and Pearly Razorfish) had their highest densities over sand. Of the 33 taxa with highest densities over rocky habitat, 17 had highest densities over moderate to high relief rock and 16 over low relief rock (Figure 13 and Table A1).

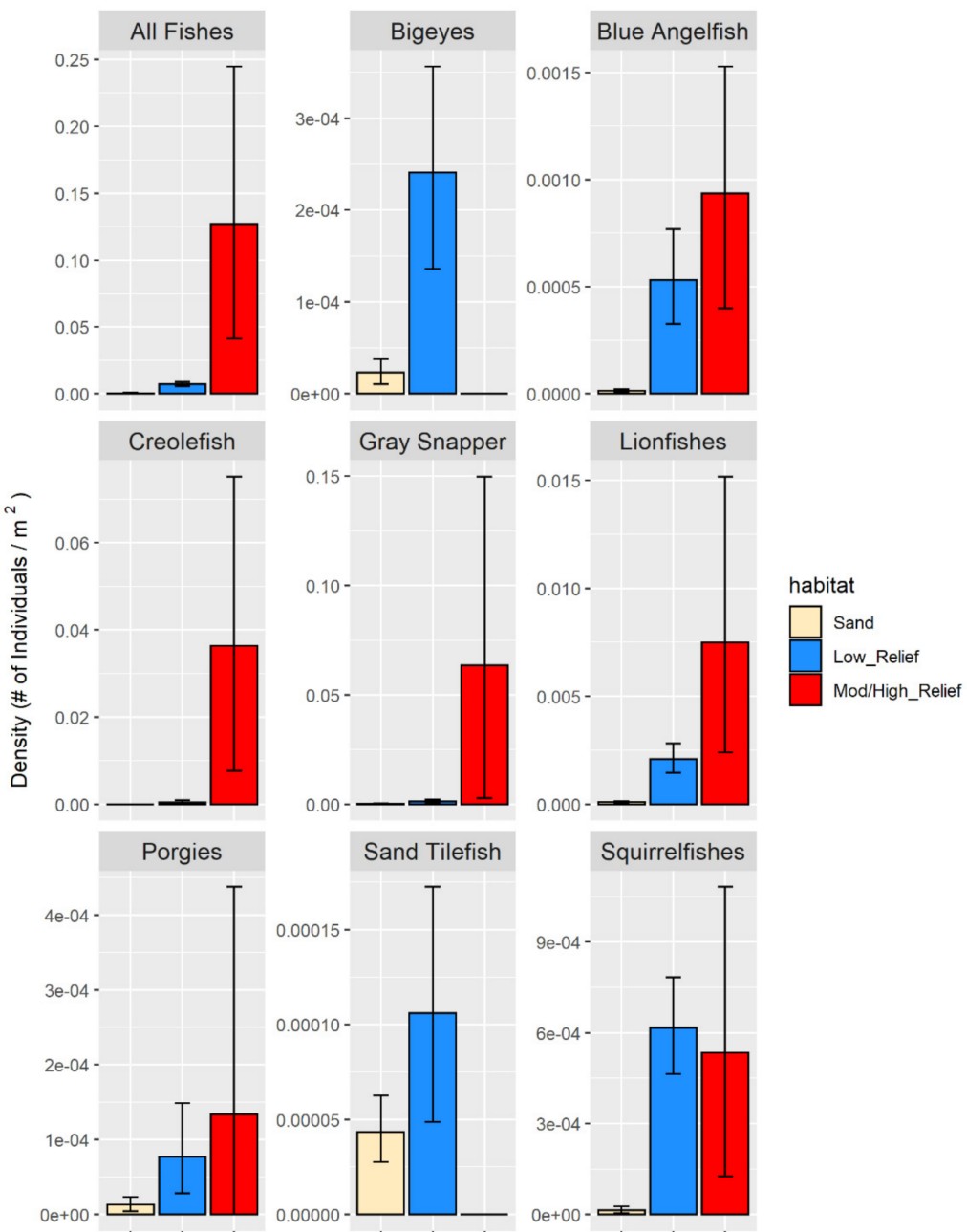

**Figure 13.** The habitat-specific densities for select fish taxa by habitat type determined from the towed video transects. Error bars represent the 95% bootstrap confidence intervals.

### 3.2.2. Multivariate Community Analysis

A total of 2030 different individual fish were observed, spanning at least 32 different species and 18 different families (Table A2) along the three observed video transects. After combining sequential 15 s bins of the same habitat into one observation and removing observations where no fish were present, there were 41 observations over sand, 80 over low relief rock, 10 over moderate relief rock, and 7 over high relief rock. The dispersions among different habitats were found not to be significantly different ($p = 0.186$), indicating that the assumption of homogeneity of multivariate dispersion was met. The PERMANOVA showed that the fish community compositions differed significantly among habitat types (Table 5, $p = 0.001$), and the pairwise comparisons found that all habitat types significantly differed from one another except between moderate and high relief rock (Table 6).

**Table 5.** Results of the PERMANOVA assessing the null hypothesis of no significant difference in fish communities among the four habitat types: sand, low relief rock, moderate relief rock, and high relief rock.

|  | Degrees of Freedom | Sum of Squares | Mean Square | F | p |
|---|---|---|---|---|---|
| **Habitat** | 3 | 4.762 | 1.58737 | 4.4454 | 0.001 |
| **Residual** | 134 | 47.849 | 0.35708 |  |  |
| **Total** | 137 | 52.611 |  |  |  |

**Table 6.** Results of the pairwise PERMANOVA tests with 9999 iterations to assess differences in fish communities between each pair of habitat types. The F statistic, a *p* value, and an adjusted *p* value using a sequential Bonferroni procedure to account for multiple comparisons are reported.

| Comparison | F | *p* | Adjusted *p* |
|---|---|---|---|
| Sand vs. Low Relief Rock | 8.685 | 0.0001 | 0.0006 |
| Sand vs. Moderate Relief Rock | 3.114 | 0.0006 | 0.0024 |
| Sand vs. High Relief Rock | 3.513 | 0.0002 | 0.0010 |
| Low Relief Rock vs. Moderate Relief Rock | 2.727 | 0.0066 | 0.0198 |
| Low Relief Rock vs. High Relief Rock | 2.410 | 0.0165 | 0.0330 |
| Moderate Relief Rock vs. High Relief Rock | 0.839 | 0.5773 | 0.5773 |

The CAP results indicated that communities over moderate and high relief rock were differentiated by having more Creolefish, Gray Snapper, Goliath Grouper, and Spanish Hogfish; low relief rocky habitats were differentiated by squirrelfishes, Blue Angelfish, lionfishes, and surgeon fishes; sand communities were differentiated by Sand Tilefish, Rainbow Runner, and remoras (Figure 14).

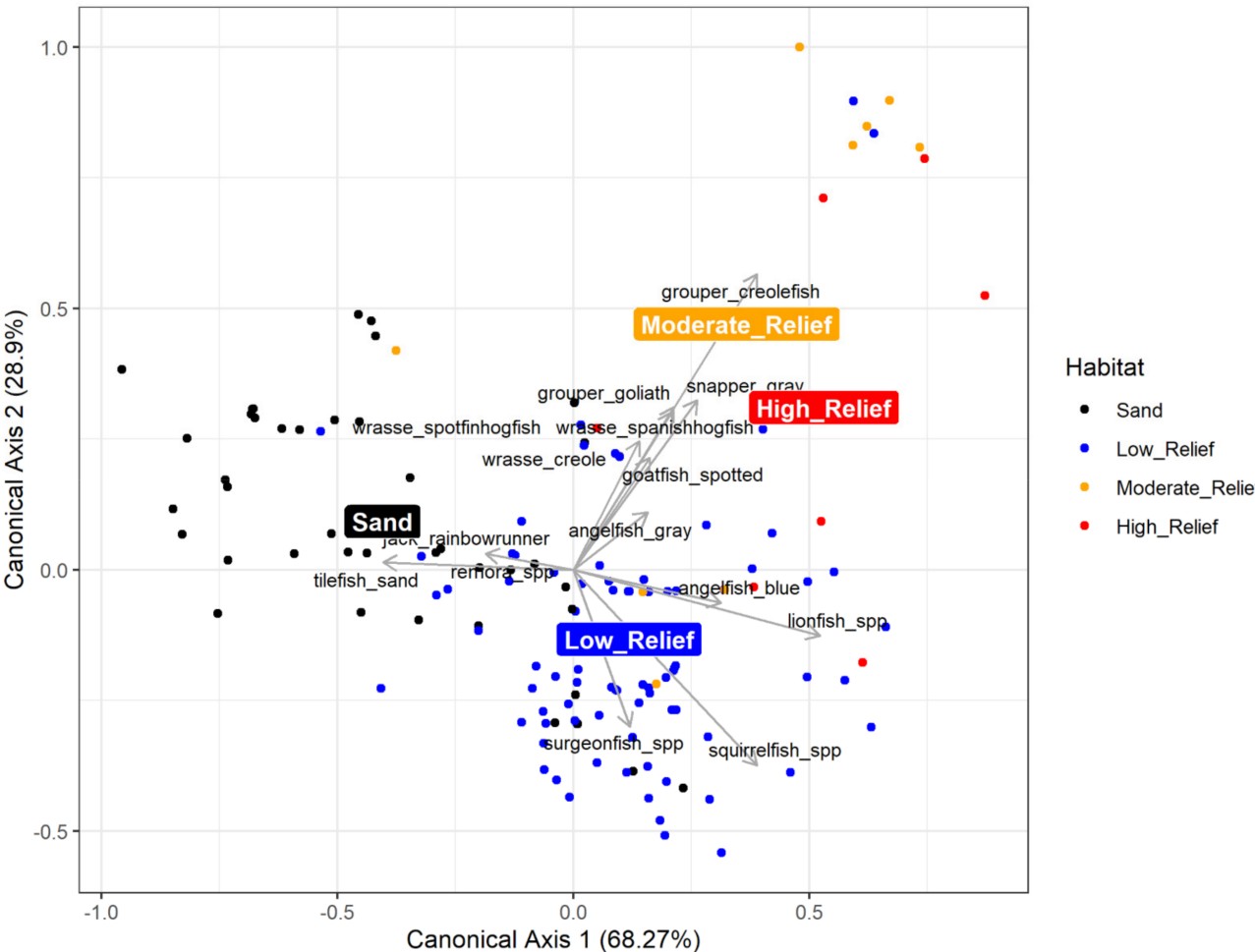

**Figure 14.** Canonical analysis of principal coordinates (CAP) plot showing the top 15 species that differentiate communities among four habitat types: sand, low relief rock, moderate relief rock, and high relief rock habitats. Points represent individual observations of a habitat type, and vectors represent the species correlation vectors. Additionally, the centroids for each habitat type are plotted.

### 3.2.3. Fish Abundance Estimates

We estimated 110,000 fish (95% CI (59842, 174961)) $\geq$15 cm in length within the 88 km$^2$ study area (Figure 15 and Table A3). Of these, 39,000 (35%) were predicted to be over sand, 25,000 (23%) over low relief rock, and 45,000 (41%) over moderate to high relief rock (Figure 15).

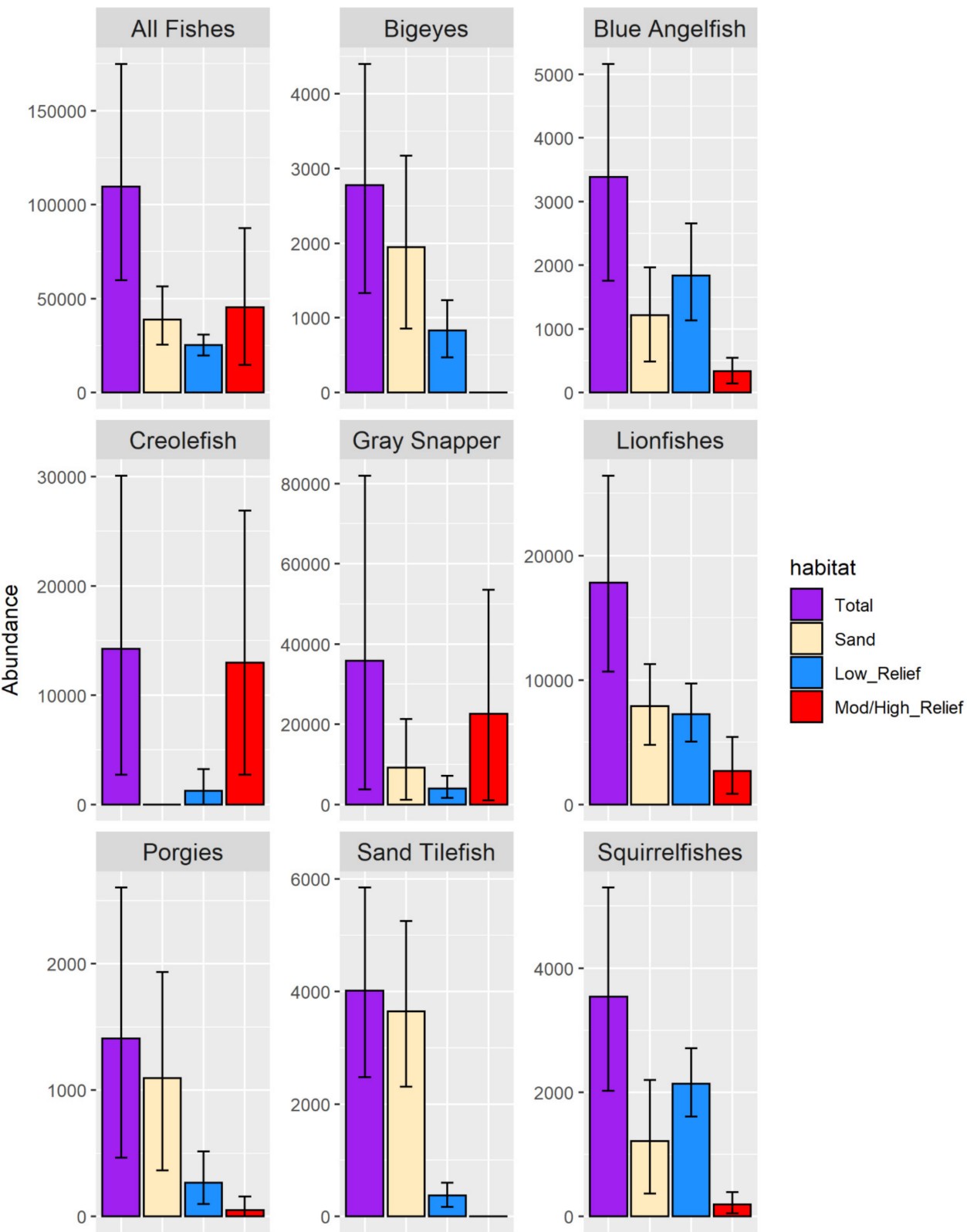

**Figure 15.** Estimates of total abundance for select fish taxa within the portion of Elbow on the west Florida shelf that was mapped using a MBES. The calculated contributions of sand, low relief rock, and moderate to high relief rock to the estimated total abundances are also shown. Extrapolations are based on the area of each habitat as determined by the combined substrate and vertical relief map using the supervised methodology (Figure 12) and the habitat-specific densities for each taxon (Table A1 and Figure 13). Error bars represent the 95% bootstrap confidence intervals.

## 4. Discussion

### 4.1. Comparison of the Supervised and Unsupervised Procedures

Both the supervised and the unsupervised map classification procedures resulted in maps of the study area with the same general configurations; both maps identified the main long rocky ridge running north to south, a smaller ridge to the west, and several small rocky outcroppings (Figures 6a and 10b). The maps looked very similar based on a visual inspection, and this was verified statistically as the maps exhibited "substantial agreement" overall and "almost perfect agreement" in terms of both spatial allocation and relative frequency of class assignment. Inspection of the difference map showed that the disagreement was mainly in transitional areas at the boundary of features. Additionally, both the supervised and the unsupervised substrate maps had very high accuracy (>97%) when predicting the validation data. However, overall accuracy can be a misleading metric when classes are highly unbalanced [90,114]. Accordingly, it is critical to evaluate maps using both user's and producer's accuracy as well as the Kappa statistic, which accounts for agreement that could occur by chance [90]. Both maps had high user's and producer's accuracies for sand. However, for rock (the rarer class), the unsupervised classification showed a moderately higher user's accuracy than did the supervised map (100% vs. 71.9%) but had a much lower producer's accuracy (35.7% vs. 82.1%). Thus, the unsupervised classification only predicted rock if it was very certain, leading to high errors of omission (false negatives). This in turn meant that the unsupervised classification was likely under-predicting the area of rock habitats, which is consistent with the finding that the supervised classification map predicted slightly more rock than the unsupervised classification map (an additional 0.08 km$^2$). Additionally, in the difference map, it was more common to see instances where the supervised classification map predicted rock and the unsupervised map predicted sand as opposed to vice versa (Figure 11), especially in areas further from the training data. This, along with the supervised model's greater performance on the validation data, indicated that the supervised model was more generalizable than the unsupervised model for predictions on new data. There are tradeoffs between overall accuracy and user's and producer's accuracy for each class. Examination of the Kappa statistic, however, indicated the supervised methodology performed better than the unsupervised methodology, and a Monte-Carlo procedure determined this difference to be significant ($p = 0.001$). In the supervised classification map, $\kappa > 0.6$, indicating "substantial agreement" between predictions and observations, while in the unsupervised classification map, $\kappa > 0.4$, indicating "moderate agreement" between predictions and observations. Although the supervised classification map performed better, the unsupervised procedure can still be useful. Particularly, the determination of acoustic clusters (Figure 10a) can be very valuable for designing the initial ground-truthing surveys, as one can make sure to sample within each acoustic cluster to maximize the likelihood that the full diversity of habitats in the area is sampled.

In addition to accuracy assessment, for the supervised methodology, we mapped the uncertainty in the classification for each individual pixel (Figure 6b). This gave us a picture of how uncertainty changed throughout the study area rather than just the average. Generally, the most uncertainty was found in rocky areas. However, some other areas also had moderate uncertainty. These other areas may represent differing morphologies of rock and sand habitats other than the ones from which the model was trained—mixed habitat classes, areas with gravel or debris, or entirely new habitats that were not observed in the video transects. This makes sense, as rock is the rarer substrate class, thus there are fewer samples upon which to train the model, and high complexity areas and habitat boundaries tend to have greater errors [56,115,116]. To improve the uncertainty within these areas, additional ground-truthing efforts would be needed to collect more observations in these areas. Assessing the uncertainty in a spatial context is very important, as the accuracy statistics calculated from the confusion matrix simply describe "average" uncertainty, but uncertainty is variable across the study area (Figure 6b) [56,117]. Despite being important, uncertainty in spatial classification of habitats is rarely assessed in benthic habitat mapping

studies [56]. The entropy map presented herein provides a method to visually depict uncertainty regardless of the number of habitat classes and can be used with other classifiers as long as the model used is capable of outputting class probabilities [57,92,93]. Additionally, another simple way to represent the (un)certainty over space is to map the probabilities of the assigned class at each pixel location or, conversely, one minus that to display the uncertainty in the classification.

### 4.2. Variable Importance for Substrate Mapping

Of the many predictor variables considered, most were either unimportant or redundant, as only 12 of the 130 predictors were retained in the final model. Seven of these predictors were derived from bathymetry and five from backscatter. The two most important predictors were slope (11 × 11) and standard deviation of backscatter (27 × 27). These two predictors had noticeably higher importance scores than the rest of the retained variables (Figure 7). Previous studies also have often found slope to be an important predictor and that including both bathymetric and backscatter information can improve classification accuracy [26,58,75,76,118]. Additionally, several predictors were found to be important at two differing spatial scales, potentially indicating that the same predictor may be representative of different properties and processes at differing scales of analysis.

### 4.3. Fish Community Analysis and Abundance Estimates

We found the fish communities to differ significantly among habitat types except between moderate and high relief rock (Tables 5 and 6). This is consistent with many other studies on the WFS and elsewhere finding both substrate and vertical relief or complexity to play important roles in shaping fish community composition [34,53,97–101,119]. In particular, our study is comparable to that by Switzer et al. [34] who also studied a portion of the Elbow, instead using baited stationary cameras and finding fish communities to differ among different types of hard-bottom habitats. In our analysis, we also identified the species driving compositional differences among habitat groups, generally finding that taxa differentiating higher relief habitats were large reef fish species (e.g., snappers and groupers), while the low relief rocky habitats were differentiated by smaller reef fish (e.g., squirrelfishes, angelfishes, and lionfishes), and sand habitats were differentiated by Sand Tilefish and some pelagic species (e.g., jacks and remoras) likely moving through the area rather than seeking rocky areas for refuge.

Overall, fish density was highest over moderate to high relief rock, followed by low relief rock and then sand (Figure 13 and Table A1), but this differed by taxa; 17 taxa had densities highest over moderate to high relief rock, 16 taxa had densities highest over low relief rock, and only 3 taxa had highest densities over sand. After extrapolating fish densities to total abundance, we estimated 110,000 fish (95% CI (59842, 174961) ≥15 cm in length within the study area (Figure 15 and Table A3). Of these, 39,000 (35%) were predicted to be over sand, 25,000 (23%) over low relief rock, and 45,000 (41%) over moderate to high relief rock (Figure 15). Rock substrate (of all relief classes) therefore contained 65% of all fish while comprising just 4% of the study area. Additionally, moderate to high relief rock made up just 0.4% of the study area but contributed more than all the surrounding sand, which covers approximately 96% of the study area. The disproportionate importance of rocky substrates, particularly moderate to high relief rocky areas, underscores the importance of offshore hard-bottom areas as "critical habitats" for demersal fish in the offshore environment. Moreover, we found that although sand habitats sustain much lower densities of fish and therefore are not likely to be targets of directed fishing efforts, they are still important to the population dynamics of species since the physical area of low density is so large. Thus, accurate assessment of these demersal reef fish populations requires sampling both sand and rock habitats; however, rock habitats, particularly those that are higher relief, should be differentially sampled with greater intensity to increase the precision of density and abundance estimates, as density and variability are greater there [120].

### 4.4. Conclusions, Limitations, and Future Work

In this study, we developed a cost-effective approach to combine limited video observations (1% of the total study area) from a towed video system with MBES data to create full-coverage substrate habitat maps and estimate the total abundance of fish taxa. The field of automated habitat mapping in the marine environment is relatively young, and there has been little agreement on optimal protocols for determining inputs or statistical methods. However, best practices have begun to emerge over the last several years [21,56,94]. This study was able to implement many of the suggestions for best practices [21,56] and did so using free open source software, which we hope will help lead to more consistent, robust, and transparent benthic habitat maps [94]. Additionally, while the analyses herein are restricted to a single study area, the methodology provides a robust analysis framework that can be applied to other areas which can be useful for survey design as the Gulf of Mexico fisheries-independent monitoring surveys transition to a design that allocates sampling effort in a way that considers habitat quantity [34,121] as well as for large scale studies that estimate total abundance of fish taxa based on habitat-specific densities [39].

The use of towed video sampling to estimate fish abundance is likely to have less bias in the "catchability" (siting probability in this context) as opposed to traditional sampling gear such as trawls [53,122]. Still, it is important to understand the biases associated with any system [123]. Paired gear catchability experiments are an obvious way to formally assess the sensitivity of these estimates to fish attractance/avoidance to towed systems and represent an important next step towards improved fish density and absolute abundance estimates using towed video systems [123]. Previous work demonstrated that most of our target species (large reef fish) exhibit neutral or weak avoidance behavior in the near-field such that these reactions generally do not prevent the fish from being identified and counted [53]; however, more data can make these findings more robust. Additionally, a more comprehensive analysis of the far-field effects (before the fish come into view of the camera) is still needed. We have upcoming work with Florida Fish and Wildlife Research Institute to do these types of analyses. Moreover, although uncertainty of the habitat map was characterized (Figure 6b), the estimates of fish abundance only considered uncertainty in the fish densities. Characterizing the uncertainty in predicted substrate represents an important step towards understanding the limitations of our study; however, more work should be done to formally incorporate this uncertainty into the abundance estimates. This can be done at the design stage by informing where more ground-truthing effort is needed in order to reduce habitat uncertainty, via comparison with other mapping approaches such as Florida Fish and Wildlife Research Institute's interpreted sidescan sonar habitat maps [35], or through novel analytical approaches that can account for errors in both terms. Additionally, addressing the impact of spatial autocorrelation using geostatistical techniques to account for the effect of space on fish densities can improve abundance estimates [124]. More work should also be done to ensure appropriate thresholds for delineating vertical relief categories, as there are high densities in higher relief areas, and therefore abundance estimates are sensitive to how these thresholds are defined. More accurate quantification of visual relief thresholds could be accomplished by accounting for the bend in the tow cable by determining an appropriate catenary factor [69] or by using acoustic positioning so that images can be more accurately projected in space onto the multibeam bathymetry [48,125] or through direct measurements of vertical relief using stereo cameras [126]. Lastly, the habitat maps and the fish densities from this study could be used to improve future towed video surveys in this area, as these data allow for the design of a truly stratified systematic sample where sampling effort is optimally allocated according to the variance in fish density estimates, which would provide more precise and unbiased estimates of abundance [120].

In addition to improved fisheries surveys, habitat maps can be valuable for more sophisticated ecological analyses in the marine environment. Evaluating biological interactions of different scales and accounting for the configuration and the heterogeneity of habitats is well established in studies of the terrestrial environment but has been less

studied in marine settings [127–129]. Improved habitat maps that incorporate analyses at multiple scales can aid in conducting these types of analyses. As habitat maps become more commonly used in conservation and management, addressing the full propagation of uncertainty throughout all analyses represents a difficult but important challenge to be addressed [94].

**Supplementary Materials:** Code for analyses is available as Rmarkdown documents and all data files necessary to run the code are provided. The following are available online at https://www.mdpi.com/article/10.3390/geosciences11040176/s1.

**Author Contributions:** Conceptualization, C.L., S.D.L. and S.A.M.; Data curation, A.R.I., J.L.B., S.E.G., M.H. and A.S.; Formal analysis, A.R.I.; Funding acquisition, C.L., S.D.L. and S.A.M.; Investigation, A.R.I., J.L.B., S.E.G., J.W.G., M.H., C.L. and A.S.; Methodology, A.R.I., J.L.B., S.E.G., S.D.L., T.S.S., A.V. and S.A.M.; Project administration, S.E.G., C.L., S.D.L. and S.A.M.; Resources, C.L. and A.S.; Software, A.R.I. and A.S.; Supervision, S.E.G., C.L., S.D.L., T.S.S. and S.A.M.; Validation, A.R.I.; Visualization, A.R.I.; Writing—original draft, A.R.I.; Writing—review & editing, A.R.I., J.L.B., S.E.G., J.W.G., M.H., S.D.L., T.S.S. and S.A.M. All authors have read and agreed to the published version of the manuscript.

**Funding:** This research was supported by the National Fish and Wildlife Foundation through the Gulf Environmental Benefit Fund (Grant 45892 to S. Murawski, C. Lembke, and S. Locker, "Restoring Fish and Sea Turtle Habitat on the West Florida Continental Shelf: Benthic Habitat Mapping, Characterization and Assessment"). Additionally, this research would not have been possible without previous work on developing the C-BASS towed camera system, which was supported by the National Oceanic and Atmospheric Administration's Advanced Sampling Technology Working Group (grant numbers NA11NMF4720284, S13-0006/P.O.AB82924) and Untrawlable Habitat Strategic Initiative (grant numbers NA10OAR4320143, NA14OAR4320260). A. Ilich was also supported by the William & Elsie Knight Endowed Fellowship Fund for Marine Science, Young Fellowship Program Fund and the Von Rosenstiel Endowed Fellowship provided by the College of Marine Science at the University of South Florida.

**Data Availability Statement:** Aside from the initial processing of the MBES surfaces, all analyses were conducted using free open source software, and all code and data necessary to run these analyses were uploaded to a Zenodo repository (https://doi.org/10.5281/zenodo.4662951, accessed on 2 April 2021). Additionally, raw and rendered versions of the R markdown documents containing code and the analytical workflow used in this study were included as a supplement. Moreover, free open source software for the calculation of GLCM textures was developed for this study (https://github.com/ailich/GLCMTextures, accessed on 2 April 2021; https://doi.org/10.5281/zenodo.4659281, accessed on 2 April 2021). Lastly, the MBES surfaces were submitted to National Centers for Environmental Information (NCEI) for permanent public archiving and will be available through their bathymetric data viewer (https://maps.ngdc.noaa.gov/viewers/bathymetry/, accessed on 2 April 2021), and any data requests can be sent by email to cscampdata@usf.edu.

**Acknowledgments:** We thank other members of the C-SCAMP group including Steve Butcher, Heather Broadbent, Edmund Hughes, and Gerardo Toro-Farmer who aided in data collection and maintenance of scientific equipment and Rachel Crabtree who created outreach materials for the project. We would also like to acknowledge the assistance of the captains and the crews of the R/V *Weatherbird II* and R/V *Bellows* operated by the Florida Institute of Oceanography in collecting these data. Additionally, we would like to thank Vincent Lecours for providing critical reviews of the manuscript and invaluable feedback on the multi-scale mapping methodology. Lastly, I would like to thank the anonymous reviewers for dedicating time to review this paper and for providing constructive feedback that helped to improve it.

**Conflicts of Interest:** The authors declare no conflict of interest.

## Appendix A

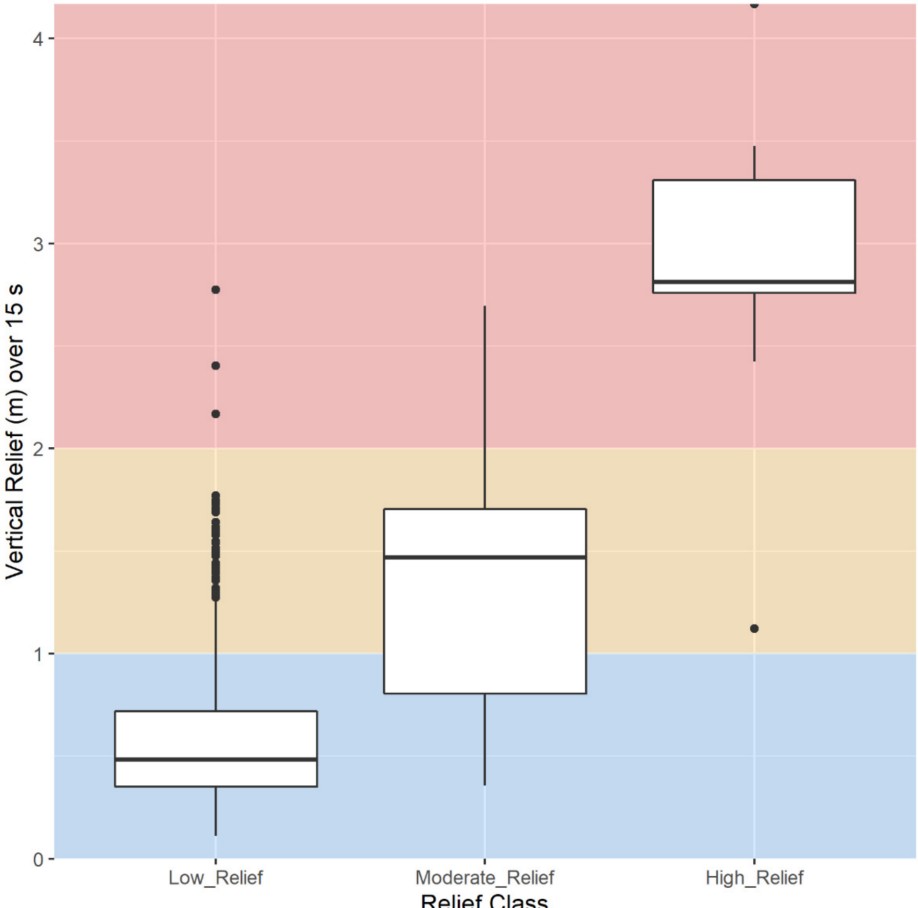

**Figure A1.** Box plot of vertical relief over 15 s time bins for each relief class. Relief was calculated from C-BASS' onboard sensors, where total depth was calculated as CBASS_Depth + CBASS_Altitude*cos(CBASS_Pitch), and the relief was calculated as the max(Total_Depth)-min(Total_Depth) for each 15 s time bin. The blue, orange, and red areas represent the range of relief values that correspond to low, medium, and high relief, respectively, based on the cutoffs provided by Smith et al. [30].

**Table A1.** Habitat-specific densities for all observed taxa in number of individuals/m$^2$ as determined from the towed video transects in the Elbow area of the west Florida shelf. Lower and upper bounds represent the 95% confidence intervals.

| Taxa | Habitat | Lower Bound | Mean | Upper Bound |
|---|---|---|---|---|
| All_Fish | Sand | 0.000305091368 | 0.000464386728 | 0.000675358543 |
| All_Fish | Low_Relief | 0.005654144932 | 0.007306868473 | 0.008924795926 |
| All_Fish | Mod/High_Relief | 0.041204024881 | 0.127203256069 | 0.245024232512 |
| amberjack_spp | Sand | 0.000000000000 | 0.000002902417 | 0.000007282130 |
| amberjack_spp | Low_Relief | 0.000038238047 | 0.000404866063 | 0.000988826791 |
| amberjack_spp | Mod/High_Relief | 0.000000000000 | 0.000133616866 | 0.000390694044 |
| angelfish_blue | Sand | 0.000005804730 | 0.000014512085 | 0.000023436164 |
| angelfish_blue | Low_Relief | 0.000326755837 | 0.000530181749 | 0.000767483569 |
| angelfish_blue | Mod/High_Relief | 0.000399268295 | 0.000935318059 | 0.001528539056 |
| angelfish_gray | Sand | 0.000000000000 | 0.000000000000 | 0.000000000000 |
| angelfish_gray | Low_Relief | 0.000009508321 | 0.000057838009 | 0.000126721804 |
| angelfish_gray | Mod/High_Relief | 0.000000000000 | 0.000133616866 | 0.000426533497 |
| angelfish_spp | Sand | 0.000000000000 | 0.000000000000 | 0.000000000000 |
| angelfish_spp | Low_Relief | 0.000000000000 | 0.000009639668 | 0.000029689030 |
| angelfish_spp | Mod/High_Relief | 0.000000000000 | 0.000000000000 | 0.000000000000 |
| bigeye_spp | Sand | 0.000010242236 | 0.000023219336 | 0.000037805805 |
| bigeye_spp | Low_Relief | 0.000136342357 | 0.000240991704 | 0.000356470592 |
| bigeye_spp | Mod/High_Relief | 0.000000000000 | 0.000000000000 | 0.000000000000 |

**Table A1.** *Cont.*

| Taxa | Habitat | Lower Bound | Mean | Upper Bound |
|---|---|---|---|---|
| boxfish_spp | Sand | 0.000000000000 | 0.000000000000 | 0.000000000000 |
| boxfish_spp | Low_Relief | 0.000000000000 | 0.000019279336 | 0.000048421997 |
| boxfish_spp | Mod/High_Relief | 0.000000000000 | 0.000000000000 | 0.000000000000 |
| butterflyfish_spp | Sand | 0.000000000000 | 0.000002902417 | 0.000007276943 |
| butterflyfish_spp | Low_Relief | 0.000009548271 | 0.000048198341 | 0.000105886188 |
| butterflyfish_spp | Mod/High_Relief | 0.000000000000 | 0.000000000000 | 0.000000000000 |
| eel_spp | Sand | 0.000000000000 | 0.000000000000 | 0.000000000000 |
| eel_spp | Low_Relief | 0.000000000000 | 0.000009639668 | 0.000029404678 |
| eel_spp | Mod/High_Relief | 0.000000000000 | 0.000000000000 | 0.000000000000 |
| goatfish_spotted | Sand | 0.000000000000 | 0.000000000000 | 0.000000000000 |
| goatfish_spotted | Low_Relief | 0.000000000000 | 0.000000000000 | 0.000000000000 |
| goatfish_spotted | Mod/High_Relief | 0.000000000000 | 0.000133616866 | 0.000430797429 |
| grouper_black | Sand | 0.000000000000 | 0.000000000000 | 0.000000000000 |
| grouper_black | Low_Relief | 0.000000000000 | 0.000000000000 | 0.000000000000 |
| grouper_black | Mod/High_Relief | 0.000000000000 | 0.000400850597 | 0.001324759414 |
| grouper_creolefish | Sand | 0.000000000000 | 0.000000000000 | 0.000000000000 |
| grouper_creolefish | Low_Relief | 0.000000000000 | 0.000356667722 | 0.000928362228 |
| grouper_creolefish | Mod/High_Relief | 0.007643028251 | 0.036343787448 | 0.075195350579 |
| grouper_gag | Sand | 0.000000000000 | 0.000001451209 | 0.000004381866 |
| grouper_gag | Low_Relief | 0.000000000000 | 0.000019279336 | 0.000059046688 |
| grouper_gag | Mod/High_Relief | 0.000000000000 | 0.000267233731 | 0.000852786071 |
| grouper_goliath | Sand | 0.000000000000 | 0.000000000000 | 0.000000000000 |
| grouper_goliath | Low_Relief | 0.000000000000 | 0.000000000000 | 0.000000000000 |
| grouper_goliath | Mod/High_Relief | 0.000000000000 | 0.000400850597 | 0.001065084821 |
| grouper_red | Sand | 0.000000000000 | 0.000001451209 | 0.000004381866 |
| grouper_red | Low_Relief | 0.000009425807 | 0.000038558673 | 0.000078463175 |
| grouper_red | Mod/High_Relief | 0.000000000000 | 0.000000000000 | 0.000000000000 |
| grouper_scamp | Sand | 0.000000000000 | 0.000000000000 | 0.000000000000 |
| grouper_scamp | Low_Relief | 0.000009680947 | 0.000144595023 | 0.000359476725 |
| grouper_scamp | Mod/High_Relief | 0.000000000000 | 0.000133616866 | 0.000430797429 |
| grouper_spp | Sand | 0.000001443556 | 0.000005804834 | 0.000011675402 |
| grouper_spp | Low_Relief | 0.000000000000 | 0.000019279336 | 0.000048726459 |
| grouper_spp | Mod/High_Relief | 0.000000000000 | 0.000133616866 | 0.000426393036 |
| jack_crevalle | Sand | 0.000000000000 | 0.000001451209 | 0.000004375663 |
| jack_crevalle | Low_Relief | 0.000000000000 | 0.000028919005 | 0.000077360701 |
| jack_crevalle | Mod/High_Relief | 0.000000000000 | 0.000000000000 | 0.000000000000 |
| jack_rainbowrunner | Sand | 0.000000000000 | 0.000002902417 | 0.000008756979 |
| jack_rainbowrunner | Low_Relief | 0.000000000000 | 0.000000000000 | 0.000000000000 |
| jack_rainbowrunner | Mod/High_Relief | 0.000000000000 | 0.000000000000 | 0.000000000000 |
| jack_spp | Sand | 0.000000000000 | 0.000000000000 | 0.000000000000 |
| jack_spp | Low_Relief | 0.000000000000 | 0.000009639668 | 0.000029677705 |
| jack_spp | Mod/High_Relief | 0.000000000000 | 0.000000000000 | 0.000000000000 |
| lionfish_spp | Sand | 0.000056869715 | 0.000094328554 | 0.000134798264 |
| lionfish_spp | Low_Relief | 0.001457130610 | 0.002091807993 | 0.002804478909 |
| lionfish_spp | Mod/High_Relief | 0.002402247287 | 0.007482544475 | 0.015170870783 |
| porgy_spp | Sand | 0.000004360488 | 0.000013060877 | 0.000023075921 |
| porgy_spp | Low_Relief | 0.000028376948 | 0.000077117345 | 0.000148411135 |
| porgy_spp | Mod/High_Relief | 0.000000000000 | 0.000133616866 | 0.000437293986 |
| remora_spp | Sand | 0.000000000000 | 0.000001451209 | 0.000004377825 |
| remora_spp | Low_Relief | 0.000000000000 | 0.000000000000 | 0.000000000000 |
| remora_spp | Mod/High_Relief | 0.000000000000 | 0.000000000000 | 0.000000000000 |
| shark_spp | Sand | 0.000000000000 | 0.000000000000 | 0.000000000000 |
| shark_spp | Low_Relief | 0.000000000000 | 0.000009639668 | 0.000029219565 |
| shark_spp | Mod/High_Relief | 0.000000000000 | 0.000000000000 | 0.000000000000 |
| snapper_gray | Sand | 0.000013061514 | 0.000108840639 | 0.000254981252 |
| snapper_gray | Low_Relief | 0.000468269192 | 0.001156760180 | 0.002055718899 |
| snapper_gray | Mod/High_Relief | 0.002742580078 | 0.063468011169 | 0.149759809054 |
| snapper_spp | Sand | 0.000000000000 | 0.000000000000 | 0.000000000000 |
| snapper_spp | Low_Relief | 0.000000000000 | 0.000019279336 | 0.000059229663 |
| snapper_spp | Mod/High_Relief | 0.000000000000 | 0.000133616866 | 0.000390694044 |
| snapper_yellowtail | Sand | 0.000000000000 | 0.000000000000 | 0.000000000000 |
| snapper_yellowtail | Low_Relief | 0.000000000000 | 0.000038558673 | 0.000116011333 |
| snapper_yellowtail | Mod/High_Relief | 0.000000000000 | 0.000267233731 | 0.000809175808 |
| squirrelfish_spp | Sand | 0.000004354164 | 0.000014512085 | 0.000026267894 |
| squirrelfish_spp | Low_Relief | 0.000464723937 | 0.000616938763 | 0.000782913741 |
| squirrelfish_spp | Mod/High_Relief | 0.000125668348 | 0.000534467462 | 0.001081371358 |
| stingray_spp | Sand | 0.000000000000 | 0.000000000000 | 0.000000000000 |
| stingray_spp | Low_Relief | 0.000000000000 | 0.000009639668 | 0.000029666787 |
| stingray_spp | Mod/High_Relief | 0.000000000000 | 0.000000000000 | 0.000000000000 |

**Table A1.** *Cont.*

| Taxa | Habitat | Lower Bound | Mean | Upper Bound |
|---|---|---|---|---|
| surgeonfish_spp | Sand | 0.000000000000 | 0.000001451209 | 0.000004388087 |
| surgeonfish_spp | Low_Relief | 0.000057776382 | 0.000134955354 | 0.000233778550 |
| surgeonfish_spp | Mod/High_Relief | 0.000000000000 | 0.000000000000 | 0.000000000000 |
| tilefish_sand | Sand | 0.000027559370 | 0.000043536256 | 0.000062686638 |
| tilefish_sand | Low_Relief | 0.000048742988 | 0.000106036350 | 0.000172505726 |
| tilefish_sand | Mod/High_Relief | 0.000000000000 | 0.000000000000 | 0.000000000000 |
| triggerfish_spp | Sand | 0.000000000000 | 0.000004353626 | 0.000010194629 |
| triggerfish_spp | Low_Relief | 0.000009284980 | 0.000038558673 | 0.000078463175 |
| triggerfish_spp | Mod/High_Relief | 0.000000000000 | 0.000000000000 | 0.000000000000 |
| wrasse_creole | Sand | 0.000000000000 | 0.000000000000 | 0.000000000000 |
| wrasse_creole | Low_Relief | 0.000000000000 | 0.000000000000 | 0.000000000000 |
| wrasse_creole | Mod/High_Relief | 0.000000000000 | 0.001336168656 | 0.004307974287 |
| wrasse_hogfish | Sand | 0.000000000000 | 0.000001451209 | 0.000004377401 |
| wrasse_hogfish | Low_Relief | 0.000009404868 | 0.000038558673 | 0.000078650541 |
| wrasse_hogfish | Mod/High_Relief | 0.000000000000 | 0.000133616866 | 0.000418370109 |
| wrasse_pearlyrazorfish | Sand | 0.000000000000 | 0.000001451209 | 0.000004378067 |
| wrasse_pearlyrazorfish | Low_Relief | 0.000000000000 | 0.000000000000 | 0.000000000000 |
| wrasse_pearlyrazorfish | Mod/High_Relief | 0.000000000000 | 0.000000000000 | 0.000000000000 |
| wrasse_spanishhogfish | Sand | 0.000000000000 | 0.000000000000 | 0.000000000000 |
| wrasse_spanishhogfish | Low_Relief | 0.000000000000 | 0.000000000000 | 0.000000000000 |
| wrasse_spanishhogfish | Mod/High_Relief | 0.000000000000 | 0.000267233731 | 0.000654899428 |
| wrasse_spotfinhogfish | Sand | 0.000000000000 | 0.000000000000 | 0.000000000000 |
| wrasse_spotfinhogfish | Low_Relief | 0.000000000000 | 0.000009639668 | 0.000029467959 |
| wrasse_spotfinhogfish | Mod/High_Relief | 0.000000000000 | 0.000133616866 | 0.000421591353 |

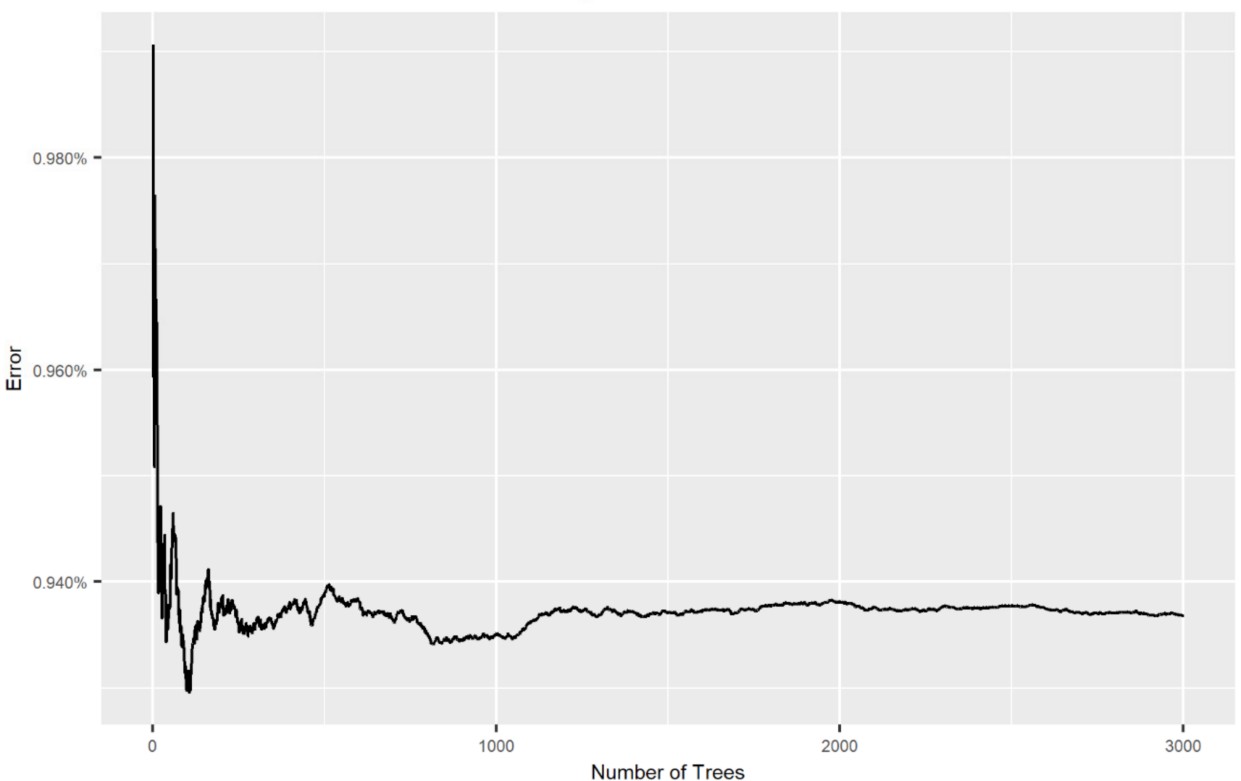

**Figure A2.** Model diagnostics showing the mean out-of-bag error as the number of decision trees in the random forest model increases. The error rate plateaus, meaning that a sufficient number of decision trees are used in the model.

**Table A2.** List of all fish taxa observed along with the number of individuals seen.

| Common Name | Scientific Name | Family | Order | Number of Individuals Observed |
|---|---|---|---|---|
| *Amberjack* spp. | *Seriola* spp. | Carangidae (Jacks) | Perciformes | 45 |
| *Angelfish* spp. | | Pomacanthidae (Angelfishes) | Perciformes | 1 |
| Blue Angelfish | *Holacanthus bermudensis* | Pomacanthidae (Angelfishes) | Perciformes | 72 |
| Gray Angelfish | *Pomacanthus arcuatus* | Pomacanthidae (Angelfishes) | Perciformes | 7 |
| *Bigeye* spp. | | Priacanthidae (Bigeyes) | Perciformes | 41 |
| *Boxfish* spp. | | Ostraciidae (Boxfishes) | Tetraodontiformes | 2 |
| Butterflyfish spp. | | Chaetodontidae (Butterflyfishes) | Perciformes | 7 |
| Eel spp. | | | Anguilliformes | 1 |
| Spotted Goatfish | *Pseudupeneus maculatus* | Mullidae (Goatfishes) | Perciformes | 1 |
| *Grouper* spp. | | Serranidae (Groupers and Sea Basses) | Perciformes | 7 |
| Black Grouper | *Mycteroperca bonaci* | Serranidae (Groupers and Sea Basses) | Perciformes | 3 |
| Atlantic Creolefish | *Paranthias furcifer* | Serranidae (Groupers and Sea Basses) | Perciformes | 309 |
| Gag Grouper | *Mycteroperca microlepis* | Serranidae (Groupers and Sea Basses) | Perciformes | 5 |
| Goliath Grouper | *Epinephelus itajara* | Serranidae (Groupers and Sea Basses) | Perciformes | 3 |
| Red Grouper | *Epinephelus morio* | Serranidae (Groupers and Sea Basses) | Perciformes | 5 |
| Scamp | *Mycteroperca phenax* | Serranidae (Groupers and Sea Basses) | Perciformes | 16 |
| *Jack* spp. | | Carangidae (Jacks) | Perciformes | 1 |
| Crevalle Jack | *Caranx hippos* | Carangidae (Jacks) | Perciformes | 4 |
| Rainbow Runner | *Elagatis bipinnulata* | Carangidae (Jacks) | Perciformes | 2 |
| *Lionfish* spp. | *Pterois* spp. | Scorpaenidae (Scorpionfishes) | Scorpaeniformes | 338 |
| *Porgy* spp. | | Sparidae (Porgies) | Perciformes | 18 |
| *Remora* spp. | | Echeneidae (Remoras) | Perciformes | 1 |
| *Shark* spp. | | | Superorder: Selachimorpha | 1 |
| *Snapper* spp. | | Lutjanidae (Snappers) | Perciformes | 3 |
| Gray Snapper | *Lutjanus griseus* | Lutjanidae (Snappers) | Perciformes | 670 |
| Yellowtail Snapper | *Ocyurus chrysurus* | Lutjanidae (Snappers) | Perciformes | 6 |
| *Squirrelfish* spp. | | Holocentridae (squirrelfishes) | Beryciformes | 78 |
| *Whiptail stingray* spp. | | Dasyatidae (Whiptail Stingrays) | Rajiformes | 1 |
| *Surgeonfish* spp. | | Acanthuridae (Surgeonfishes) | Perciformes | 15 |

**Table A2.** *Cont.*

| Common Name | Scientific Name | Family | Order | Number of Individuals Observed |
|---|---|---|---|---|
| Sand Tilefish | *Malacanthus plumieri* | Malacanthidae (tilefishes) | Perciformes | 41 |
| *Triggerfish* spp. | | Balistidae (Triggerfishes) | Tetraodontiformes | 7 |
| Creole Wrasse | *Clepticus parrae* | Labridae (Wrasses) | Perciformes | 10 |
| Hogfish | *Lachnolaimus maximus* | Labridae (Wrasses) | Perciformes | 6 |
| Pearly Razorfish | *Xyrichtys novacula* | Labridae (Wrasses) | Perciformes | 1 |
| Spanish Hogfish | *Bodianus rufus* | Labridae (Wrasses) | Perciformes | 2 |
| Spotfin Hogfish | *Bodianus pulchellus* | Labridae (Wrasses) | Perciformes | 2 |
| Large No ID | | | | 298 |

**Table A3.** Estimates of total abundance for all observed taxa within the portion of Elbow on the west Florida shelf that was mapped using a MBES. The calculated contributions of each habitat type (sand, low relief rock, and moderate to high relief rock) to the estimated total abundance are also shown. Extrapolations are based on the habitat-specific densities (Table A1 and Figure 13), and the area of each habitat based on the combined substrate and relief map (Figure 12). Upper and lower bounds represent the 95% bootstrap confidence intervals.

| Taxa | Habitat | Lower Bound | Mean | Upper Bound |
|---|---|---|---|---|
| All_Fish | Total | 59,842 | 109,616 | 174,961 |
| All_Fish | Sand | 25,552 | 38,893 | 56,563 |
| All_Fish | Low_Relief | 19,576 | 25,299 | 30,900 |
| All_Fish | Mod/High_Relief | 14,714 | 45,424 | 87,498 |
| amberjack_spp | Total | 132 | 1693 | 4173 |
| amberjack_spp | Sand | 0 | 243 | 610 |
| amberjack_spp | Low_Relief | 132 | 1402 | 3424 |
| amberjack_spp | Mod/High_Relief | 0 | 48 | 140 |
| angelfish_blue | Total | 1760 | 3385 | 5166 |
| angelfish_blue | Sand | 486 | 1215 | 1963 |
| angelfish_blue | Low_Relief | 1131 | 1836 | 2657 |
| angelfish_blue | Mod/High_Relief | 143 | 334 | 546 |
| angelfish_gray | Total | 33 | 248 | 591 |
| angelfish_gray | Sand | 0 | 0 | 0 |
| angelfish_gray | Low_Relief | 33 | 200 | 439 |
| angelfish_gray | Mod/High_Relief | 0 | 48 | 152 |
| angelfish_spp | Total | 0 | 33 | 103 |
| angelfish_spp | Sand | 0 | 0 | 0 |
| angelfish_spp | Low_Relief | 0 | 33 | 103 |
| angelfish_spp | Mod/High_Relief | 0 | 0 | 0 |
| bigeye_spp | Total | 1330 | 2779 | 4401 |
| bigeye_spp | Sand | 858 | 1945 | 3166 |
| bigeye_spp | Low_Relief | 472 | 834 | 1234 |
| bigeye_spp | Mod/High_Relief | 0 | 0 | 0 |
| boxfish_spp | Total | 0 | 67 | 168 |
| boxfish_spp | Sand | 0 | 0 | 0 |
| boxfish_spp | Low_Relief | 0 | 67 | 168 |
| boxfish_spp | Mod/High_Relief | 0 | 0 | 0 |
| butterflyfish_spp | Total | 33 | 410 | 976 |
| butterflyfish_spp | Sand | 0 | 243 | 609 |
| butterflyfish_spp | Low_Relief | 33 | 167 | 367 |
| butterflyfish_spp | Mod/High_Relief | 0 | 0 | 0 |

**Table A3.** *Cont.*

| Taxa | Habitat | Lower Bound | Mean | Upper Bound |
|---|---|---|---|---|
| eel_spp | Total | 0 | 33 | 102 |
| eel_spp | Sand | 0 | 0 | 0 |
| eel_spp | Low_Relief | 0 | 33 | 102 |
| eel_spp | Mod/High_Relief | 0 | 0 | 0 |
| goatfish_spotted | Total | 0 | 48 | 154 |
| goatfish_spotted | Sand | 0 | 0 | 0 |
| goatfish_spotted | Low_Relief | 0 | 0 | 0 |
| goatfish_spotted | Mod/High_Relief | 0 | 48 | 154 |
| grouper_black | Total | 0 | 143 | 473 |
| grouper_black | Sand | 0 | 0 | 0 |
| grouper_black | Low_Relief | 0 | 0 | 0 |
| grouper_black | Mod/High_Relief | 0 | 143 | 473 |
| grouper_creolefish | Total | 2729 | 14,213 | 30,067 |
| grouper_creolefish | Sand | 0 | 0 | 0 |
| grouper_creolefish | Low_Relief | 0 | 1235 | 3214 |
| grouper_creolefish | Mod/High_Relief | 2729 | 12,978 | 26,852 |
| grouper_gag | Total | 0 | 284 | 876 |
| grouper_gag | Sand | 0 | 122 | 367 |
| grouper_gag | Low_Relief | 0 | 67 | 204 |
| grouper_gag | Mod/High_Relief | 0 | 95 | 305 |
| grouper_goliath | Total | 0 | 143 | 380 |
| grouper_goliath | Sand | 0 | 0 | 0 |
| grouper_goliath | Low_Relief | 0 | 0 | 0 |
| grouper_goliath | Mod/High_Relief | 0 | 143 | 380 |
| grouper_red | Total | 33 | 255 | 639 |
| grouper_red | Sand | 0 | 122 | 367 |
| grouper_red | Low_Relief | 33 | 134 | 272 |
| grouper_red | Mod/High_Relief | 0 | 0 | 0 |
| grouper_scamp | Total | 34 | 548 | 1398 |
| grouper_scamp | Sand | 0 | 0 | 0 |
| grouper_scamp | Low_Relief | 34 | 501 | 1245 |
| grouper_scamp | Mod/High_Relief | 0 | 48 | 154 |
| grouper_spp | Total | 121 | 601 | 1299 |
| grouper_spp | Sand | 121 | 486 | 978 |
| grouper_spp | Low_Relief | 0 | 67 | 169 |
| grouper_spp | Mod/High_Relief | 0 | 48 | 152 |
| jack_crevalle | Total | 0 | 222 | 634 |
| jack_crevalle | Sand | 0 | 122 | 366 |
| jack_crevalle | Low_Relief | 0 | 100 | 268 |
| jack_crevalle | Mod/High_Relief | 0 | 0 | 0 |
| jack_rainbowrunner | Total | 0 | 243 | 733 |
| jack_rainbowrunner | Sand | 0 | 243 | 733 |
| jack_rainbowrunner | Low_Relief | 0 | 0 | 0 |
| jack_rainbowrunner | Mod/High_Relief | 0 | 0 | 0 |
| jack_spp | Total | 0 | 33 | 103 |
| jack_spp | Sand | 0 | 0 | 0 |
| jack_spp | Low_Relief | 0 | 33 | 103 |
| jack_spp | Mod/High_Relief | 0 | 0 | 0 |
| lionfish_spp | Total | 10,666 | 17,815 | 26,417 |
| lionfish_spp | Sand | 4763 | 7900 | 11,290 |
| lionfish_spp | Low_Relief | 5045 | 7242 | 9710 |
| lionfish_spp | Mod/High_Relief | 858 | 2672 | 5418 |
| porgy_spp | Total | 463 | 1409 | 2603 |
| porgy_spp | Sand | 365 | 1094 | 1933 |
| porgy_spp | Low_Relief | 98 | 267 | 514 |
| porgy_spp | Mod/High_Relief | 0 | 48 | 156 |
| remora_spp | Total | 0 | 122 | 367 |
| remora_spp | Sand | 0 | 122 | 367 |

**Table A3.** *Cont.*

| Taxa | Habitat | Lower Bound | Mean | Upper Bound |
|---|---|---|---|---|
| remora_spp | Low_Relief | 0 | 0 | 0 |
| remora_spp | Mod/High_Relief | 0 | 0 | 0 |
| shark_spp | Total | 0 | 33 | 101 |
| shark_spp | Sand | 0 | 0 | 0 |
| shark_spp | Low_Relief | 0 | 33 | 101 |
| shark_spp | Mod/High_Relief | 0 | 0 | 0 |
| snapper_gray | Total | 3695 | 35,785 | 81,952 |
| snapper_gray | Sand | 1094 | 9116 | 21,355 |
| snapper_gray | Low_Relief | 1621 | 4005 | 7118 |
| snapper_gray | Mod/High_Relief | 979 | 22,664 | 53,479 |
| snapper_spp | Total | 0 | 114 | 345 |
| snapper_spp | Sand | 0 | 0 | 0 |
| snapper_spp | Low_Relief | 0 | 67 | 205 |
| snapper_spp | Mod/High_Relief | 0 | 48 | 140 |
| snapper_yellowtail | Total | 0 | 229 | 691 |
| snapper_yellowtail | Sand | 0 | 0 | 0 |
| snapper_yellowtail | Low_Relief | 0 | 134 | 402 |
| snapper_yellowtail | Mod/High_Relief | 0 | 95 | 289 |
| squirrelfish_spp | Total | 2019 | 3542 | 5297 |
| squirrelfish_spp | Sand | 365 | 1215 | 2200 |
| squirrelfish_spp | Low_Relief | 1609 | 2136 | 2711 |
| squirrelfish_spp | Mod/High_Relief | 45 | 191 | 386 |
| stingray_spp | Total | 0 | 33 | 103 |
| stingray_spp | Sand | 0 | 0 | 0 |
| stingray_spp | Low_Relief | 0 | 33 | 103 |
| stingray_spp | Mod/High_Relief | 0 | 0 | 0 |
| surgeonfish_spp | Total | 200 | 589 | 1177 |
| surgeonfish_spp | Sand | 0 | 122 | 368 |
| surgeonfish_spp | Low_Relief | 200 | 467 | 809 |
| surgeonfish_spp | Mod/High_Relief | 0 | 0 | 0 |
| tilefish_sand | Total | 2477 | 4013 | 5847 |
| tilefish_sand | Sand | 2308 | 3646 | 5250 |
| tilefish_sand | Low_Relief | 169 | 367 | 597 |
| tilefish_sand | Mod/High_Relief | 0 | 0 | 0 |
| triggerfish_spp | Total | 32 | 498 | 1125 |
| triggerfish_spp | Sand | 0 | 365 | 854 |
| triggerfish_spp | Low_Relief | 32 | 134 | 272 |
| triggerfish_spp | Mod/High_Relief | 0 | 0 | 0 |
| wrasse_creole | Total | 0 | 477 | 1538 |
| wrasse_creole | Sand | 0 | 0 | 0 |
| wrasse_creole | Low_Relief | 0 | 0 | 0 |
| wrasse_creole | Mod/High_Relief | 0 | 477 | 1538 |
| wrasse_hogfish | Total | 33 | 303 | 788 |
| wrasse_hogfish | Sand | 0 | 122 | 367 |
| wrasse_hogfish | Low_Relief | 33 | 134 | 272 |
| wrasse_hogfish | Mod/High_Relief | 0 | 48 | 149 |
| wrasse_pearlyrazorfish | Total | 0 | 122 | 367 |
| wrasse_pearlyrazorfish | Sand | 0 | 122 | 367 |
| wrasse_pearlyrazorfish | Low_Relief | 0 | 0 | 0 |
| wrasse_pearlyrazorfish | Mod/High_Relief | 0 | 0 | 0 |
| wrasse_spanishhogfish | Total | 0 | 95 | 234 |
| wrasse_spanishhogfish | Sand | 0 | 0 | 0 |
| wrasse_spanishhogfish | Low_Relief | 0 | 0 | 0 |
| wrasse_spanishhogfish | Mod/High_Relief | 0 | 95 | 234 |
| wrasse_spotfinhogfish | Total | 0 | 81 | 253 |
| wrasse_spotfinhogfish | Sand | 0 | 0 | 0 |
| wrasse_spotfinhogfish | Low_Relief | 0 | 33 | 102 |
| wrasse_spotfinhogfish | Mod/High_Relief | 0 | 48 | 151 |

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
