# Peer review of "Integrating Towed Underwater Video and Multibeam Acoustics for Marine Benthic Habitat Mapping and Fish Population Estimation"

_geosciences, doi:10.3390/geosciences11040176_

Round 1

Reviewer 1 Report

The authors do a very thorough job of developing an approach to use a combination of video and sonar data to generate a wide-area assessment of benthic habitat and then use this assessment to estimate fish abundance. I think the paper is very well-written and the authors provide a careful and clear statistical assessment of their approach, there are a couple points that should be addressed prior to publication:
  • While I believe the authors do a very thorough job of developing a technique to use video data to serve as ground truth for use in the inversion of multibeam data for habitat, I'm less convinced by the fish abundance analysis. There is no real "ground truth" to determine whether this result is correct. Is there any independent data that can be used to validate this estimate of fish abundance? Are there other studies of The Elbow that can be leveraged to support your results? At the very least, the authors should suggest a way that they could validate this result.

  • I think the entropy/uncertainty map given in Figure 5b is an important data product in this study and as the authors point out, this type of uncertainty is rarely assessed in benthic mapping studies (but, in my opinion, should be.) But while the authors have taken an important first step by providing an uncertainty assessment, the next question that needs to be addressed, in both this study and for the broader fisheries community, is how should this information be used?

While the authors use the results of the supervised map classification procedure to estimate the fish abundance and provide estimates of the abundance uncertainty based on the statistics of the fish observations, they ignore the potential impacts of the habitat uncertainty. This uncertainty will obviously propagate into the fish abundance. It isn't immediately clear to me how that might be done (or even if it can be at this stage), but the authors should at least acknowledge that there should be an impact on the abundance uncertainties and if possible, discuss how one might go about incorporating these uncertainties.

  • While the authors conclude that he supervised classification performs better than the unsupervised classification, the correspondence between the Shannon entropy map (Fig. 5b) and the Acoustic Cluster map (Fig. 9) suggests how the unsupervised classification might still be of value for the sonar/video technique developed in this paper. Given that the sonar and the video surveys were conducted several months apart, one could have used the cluster map to plan the camera tows such that it would pass through all of the cluster areas which would have, in retrospect, also passed through the areas of high uncertainty. This type of procedure might be worth mentioning in the discussion, or at the very least, this is something the authors might want to consider as they continue to develop and apply this technique.  

Reviewer 2 Report

I found this to be an excellent investigation with important goals for habitat and fisheries researchers. The methods were sound and the findings well supported. It should be published. 

The ms was generally well written. One concern I had was the detailed explanation of the fundamentals of the statistical / modeling approaches. I think the descriptions of some of the statistical applications should be slimmed down.  I am saying this as one who is mostly interested in knowing the outcome of comparative analyses so that I have trust in the ecological findings.

On the other hand, for an acoustics habitat survey modeler the detail rationale and description of survey methodology is probably of primary interest. GEOHAB embraces the importance of survey methodologies and technologies in relation to ecology.

Therefore, I suggest one of two strategies:

1) remove some of the extraneous text which is not necessary for a peer-reviewed publication.  Trim the ms down and publish it.

or

2) turn this into two papers. One focused on the methods including statistical treatment of the data. And a follow up paper focusing on the utility of this approach for broad-scale habitat and fish surveys.

Personally I would prefer option #2.  I imagine that the authors may see this as more work than necessary, but frankly I am not so sure that it would be all that much work to separate the methods from the benefits to surveys (with the fish- habitat correlations), and it would be kinder to the readers.  Also, the entry-level scientists would likely appreciate two publications rather than one in their CVs. 

Please also see the manuscript (attached) with comments. 

Reviewer 3 Report

Very nice paper, appropriate research design and comprehensively described methods. I would like to see this paper published after considering some following comments: 

General comments:

The article needs the formulation of a proper research question/hypothesis. a good place to complete this part is the end of the introduction. Then, refer to the research question/hypothesis at the beginning of the discussion.

Moreover, I suggest to organize the structure of discussion by separating paragraphs for at least some of classical sub-sections, like e.g.: reference to the main research objectives, summary of the main findings of the article, interpretation of the main findings, limitations of your research, recommendations for future research.

The manuscript's minor drawbacks are incorrect references (errors) to figure captions and/or other parts. 

Some specific comments:

line 50: consider the extension of the description by Synthetic Aperture Sonar that have enhanced measurement capabilities than simple sidescan

line 61: because extraction of secondary derivatives of mbes datasets is a topic of particular interest of the scientific community, I suggest supplementing this part with other types of derivatives, like textural parameters of backscatter (Montereale-Gavazzi et al., 2017; Prampolini et al., 2018), as well as spectral parameters of bathymetry (Trzcinska et al., 2020).

  • Montereale-Gavazzi, G.; Roche, M.; Lurton, X.; Degrendele, K.; Terseleer, N.; Van Lancker, V. Seafloor change detection using multibeam echosounder backscatter: case study on the Belgian part of the North Sea. Marine Geophysical Research 2018, 39, 229-247, doi:10.1007/s11001-017-9323-6.
  • Prampolini, M.; Blondel, P.; Foglini, F.; Madricardo, F. Habitat mapping of the Maltese continental shelf using acoustic textures and bathymetric analyses. Estuarine, Coastal and Shelf Science 2018, 207, 483-498, doi:10.1016/j.ecss.2017.06.002.
  • Trzcinska, K.; Janowski, L.; Nowak, J.; Rucinska-Zjadacz, M.; Kruss, A.; Schneider von Deimling, J.; Pocwiardowski, P.; Tegowski, J. Spectral features of dual-frequency multibeam echosounder data for benthic habitat mapping. Marine Geology 2020, 427, 106239, doi:10.1016/j.margeo.2020.106239.

line 111: since IHO S-44 provides minimum requirements for surveying, I recommend supplementing specific properties of your survey.

line 200: vague. What do you mean by acoustic signature?

line 291: if you mention that the random forest model was implemented in 'a number of previous benthic habitat mapping studies', it would be beneficial to cite more than one reference.
